# Calcium/calmodulin-dependent protein kinase IV promotes imiquimod-induced psoriatic inflammation via macrophages and keratinocytes in mice

Liang Yong [1,2,3,4,5,8], Yafen Yu[1,2,3,4,5,8], Bao Li[6,8], Huiyao Ge[1,2,3,4,5], Qi Zhen[1,2,3,4,5], Yiwen Mao[1,2,3,4,5], Yanxia Yu[1,2,3,4,5], Lu Cao[1,2,3,4,5], Ruixue Zhang[1,2,3,4,5], Zhuo Li[1,2,3,4,5], Yirui Wang[1,2,3,4,5], Wencheng Fan[1,2,3,4,5], Chang Zhang[1,2,3,4,5], Daiyue Wang[1,2,3,4,5], Sihan Luo[1,2,3,4,5], Yuanming Bai[1,2,3,4,5], Shirui Chen[1,2,3,4,5], Weiwei Chen[1,2,3,4,5], Miao Liu[7], Jijia Shen[7] & Liangdan Sun [1,2,3,4,5] ✉

CaMK4 has an important function in autoimmune diseases, and the contribution of CaMK4 in psoriasis remains obscure. Here, we show that CaMK4 expression is significantly increased in psoriatic lesional skin from psoriasis patients compared to healthy human skin as well as inflamed skin from an imiquimod (IMQ)-induced mouse model of psoriasis compared to healthy mouse skin. *Camk4*-deficient (*Camk4*$^{-/-}$) mice treated with IMQ exhibit reduced severity of psoriasis compared to wild-type (WT) mice. There are more macrophages and fewer IL-17A$^+$γδ TCR$^+$ cells in the skin of IMQ-treated *Camk4*$^{-/-}$ mice compared to IMQ-treated WT mice. CaMK4 inhibits IL-10 production by macrophages, thus allowing excessive psoriatic inflammation. Deletion of *Camk4* in macrophages alleviates IMQ-induced psoriatic inflammation in mice. In keratinocytes, CaMK4 inhibits apoptosis as well as promotes cell proliferation and the expression of pro-inflammatory genes such as *S100A8* and *CAMP*. Taken together, these data indicate that CaMK4 regulates IMQ-induced psoriasis by sustaining inflammation and provides a potential target for psoriasis treatment.

Psoriasis is a chronic and immune-mediated disease associated with environmental and genetic factors that affect 2–3% of the worldwide population[1,2]. The innate and adaptive immune systems are both involved in the pathogenesis of psoriasis, and myeloid cells, T cells, and keratinocytes (KCs) have key functions in the process[3–6]. The dendritic cell (DC)/IL-23/T17 cell/IL-17/KC axis is a critical constituent of a positive feedback loop that promotes psoriasis progression[3–6].

The skin, which consists of the epidermis and dermis, is the first line of defense against the external environment. Various myeloid cells, including Langerhans cells (LCs), DCs, macrophages, monocytes, and neutrophils, participate in skin innate immunity[7,8]. Conventional DCs (cDCs) and monocyte-derived DCs (moDCs) drive psoriatic inflammation, whereas plasmacytoid DCs (pDCs) are dispensable for IMQ-induced psoriatic plaque formation[9–12]. Macrophages in the skin

[1]Department of Dermatology, the First Affiliated Hospital of Anhui Medical University, Hefei, China. [2]Institute of Dermatology, Anhui Medical University, Hefei, China. [3]Key Laboratory of Dermatology (Anhui Medical University), Ministry of Education, Hefei, China. [4]Inflammation and Immune Mediated Diseases Laboratory of Anhui Province, Hefei, China. [5]Anhui Provincial Institute of Translational Medicine, Hefei, China. [6]Integrated Laboratory, School of Basic Medical Sciences, Anhui Medical University, Hefei, China. [7]Anhui Provincial Laboratory of Microbiology and Parasitology; Department of Microbiology and Parasitology, Anhui Medical University, Hefei, China. [8]These authors contributed equally: Liang Yong, Yafen Yu, Bao Li. ✉e-mail: ahmusld@163.com

are located in the dermis; they have scavenging and phagocytic activities as well as anti-inflammatory properties that contribute to the destruction of microorganisms and the maintenance of skin homeostasis[7,11,13]. Classically activated macrophages (M1 macrophages) and alternatively activated macrophages (M2 macrophages) undergo phenotypic switching according to the tissue microenvironment. M2 macrophages contribute to wound healing, angiogenesis, and tissue remodeling[14–16].

Cytokines derived from T cells and antigen-presenting cells act on KCs to contribute to the histological phenotype of psoriasis. For example, TNF-stimulated KCs show upregulated expression of genes involved in immune and inflammatory responses, tissue remodeling, cell motility, cell cycle, and apoptosis[17]. In addition, the IL-22 receptor is highly expressed in KCs, and the expression of β-defensin 2 and 3 is increased in IL-22-treated KCs[18]. KCs are also the main cell type expressing the IL-17 receptor in psoriasis, and IL-17A increases the expression of pro-inflammatory cytokines, chemokines, and anti-microbial peptides (AMPs) in KCs[19,20]. In fact, the combination of TNF and IL-17A exerts a synergistic effect that induces greater changes in gene expression than either factor alone[21]. Collectively, these cytokines potentiate the hyperproliferation of KCs and increase the expression of cytokines, chemokines, and AMPs. This positive feedback loop acts on DCs, T cells, and neutrophils to perpetuate psoriatic inflammation.

Calcium/calmodulin-dependent protein kinase IV (CaMK4), which belongs to the serine/threonine kinase family, regulates the expression of several genes, including IL-2 and IL-17[22–24]. CaMK4 is increased in T cells of patients with systemic lupus erythematosus (SLE) and is required for Th17 cell differentiation[24]. Moreover, pharmacologic inhibition of CaMK4 decreases IL-17 transcription by downregulating the AKT/mTOR/S6K/RORγt and cAMP response element modulator α (CREM-α) pathways[24]. Furthermore, genetic deletion or pharmacologic inhibition of CaMK4 ameliorates Th17 cell-mediated autoimmune diseases, such as experimental autoimmune encephalomyelitis (EAE) and lupus-like disease in mice[24–26]. Psoriasis is also an IL-17-associated autoimmune disease, and the mechanism by which CaMK4 affects psoriasis remains unknown.

Here, we demonstrate that CaMK4 expression is significantly increased in psoriatic lesional skin from patients with psoriasis and mice with IMQ-induced psoriasis compared to healthy skin. Genetic deletion of CaMK4 effectively alleviates IMQ-induced psoriatic inflammation, and this alleviation is accompanied by decreased skin IL-17A-producing T cells, including CD4+ and γδ TCR+ cells. In addition to the reported effect of CaMK4 on IL-17A production, we also document that CaMK4 has functions in dermal macrophages and epidermal KCs to affect psoriasis progression. In dermal macrophages, CaMK4 inhibits IL-10 production through downregulating Erk1/2 and p38 phosphorylation. In epidermal KCs, the CaMK4-AKT-NF-κB pathway promotes pro-inflammatory phenotypes. Understanding the mechanism by which CaMK4 affects the pathogenesis of psoriasis may lead to the development of therapeutic strategies for psoriasis.

## Results

### CaMK4 expression is increased in psoriatic lesional skin from patients with psoriasis and mice with IMQ-induced psoriasis

To better understand the contribution of CaMK4 in psoriasis, we first detected CaMK4 expression in patients with psoriasis and mice with IMQ-induced psoriasis by various methods. Immunohistochemical analysis of human skin tissues showed that the number of CaMK4+ cells in both the epidermis and dermis of psoriatic lesional skin was higher than that of non-lesional skin and healthy skin (Fig. 1a). Meanwhile, an increased number of CaMK4+ cells in the epidermis and dermis of psoriatic non-lesional skin was observed compared to healthy skin (Fig. 1a). Additionally, we observed an increased number of CaMK4+ macrophages in psoriatic lesional skin compared to healthy skin

(Supplementary Fig. 1a). The mRNA level of *CAMK4* was also higher in the cells of peripheral blood of patients with psoriasis than in that of healthy controls (Fig. 1b). We then identified CaMK4 expression in human peripheral immune cell subsets by flow cytometry. The gating strategy for the identification of human peripheral immune cell subsets was shown in Supplementary Fig. 1b. We found that the proportion of CaMK4+ cells and the mean fluorescence intensity (MFI) of CaMK4 were both higher in peripheral T cell subsets (CD4+, CD8+, and double-negative (DN) cells) and CD14+ monocytes from patients with psoriasis than in those from healthy controls (Fig. 1c and Supplementary Fig. 1c). However, CaMK4 expression was higher in CD14+ monocytes from both patients with psoriasis and healthy controls than in T cell subsets (Fig. 1c). We subsequently analyzed the relationships between the expression of pro-inflammatory cytokines and *CAMK4* expression in the cells of peripheral blood from patients with psoriasis using correlation analysis. The expression of *IL1B* and *IL12B* was positively correlated with *CAMK4* expression (Fig. 1d).

We also found that CaMK4 expression was upregulated in IMQ-treated mouse skin compared to healthy mouse skin by immunohistochemical analysis (Fig. 1e). It has been reported that CaMK4 expression and activity are increased in T cells in patients with SLE. Activated CaMK4 mitigates IL-2 transcription by phosphorylating CREM-α, whereas it promotes IL-17 production through the AKT/mTOR/S6K/RORγt and CREM-α pathways in T cells of SLE[24,27]. Our results also showed that CaMK4 expression was higher in T cells from IMQ-treated mouse skin than in those from healthy mouse skin (Supplementary Fig. 2). In addition, we found that CaMK4 expression was increased in F4/80+ cells, CD11c+ cells, and primary KCs but not in B cells and neutrophils from IMQ-treated mouse skin compared to healthy mouse skin (Fig. 1f, g, Supplementary Fig. 2), suggesting that CaMK4 may function in myeloid cells and KCs to influence IMQ-induced psoriasis.

### *Camk4* deficiency alleviates the severity of IMQ-induced psoriasis

Then, we established a classic IMQ-induced psoriasis mouse model using *Camk4*−/− and WT (*Camk4*+/+) mice. The severity scoring and histological analysis showed that the severity of psoriasis was alleviated in IMQ-treated *Camk4*−/− mice compared to IMQ-treated *Camk4*+/+ mice (Fig. 2a–d). To obtain a comprehensive overview of the transcriptional signature in IMQ-treated *Camk4*+/+ and *Camk4*−/− mice, we performed RNA sequencing of the whole skin. Differentially expressed genes in the skin between IMQ-treated *Camk4*+/+ and *Camk4*−/− mice included those associated with inflammatory response, chemokine signaling pathway, cytokine-cytokine receptor interaction, and Fc gamma R-mediated phagocytosis (Supplementary Fig. 3a, b). In addition, gene sets known to be affected in psoriasis were highlighted, including cytokines (*Il17a*, *Il17f*, *Il18*, *Il20*, and *Il22*), chemokines (*Ccl20*, *Ccl27a*, and *Ccl27b*), AMPs (*S100a8* and *S100a9*), epidermis development (*Lce3f*, *Sprr2b*, and *Krt15*), tissue repair-related genes (*Il10ra*, *Mmp2*, *Mmp9*, *Hgf*, *Igf1*, and *Tgfb1*), and phagocytosis-related genes (*Trem2*, *Gpnmb*, *Cd5l*, and *Fcrls*) (Fig. 2e and Supplementary Fig. 3c). We performed quantitative PCR to confirm the RNA sequencing data. The quantitative PCR results were consistent with our RNA sequencing analysis and showed significant reductions of *S100a8*, *S100a9*, *Il1b*, *Il17a*, *Il17f*, *Il22*, and *Ccl20* in the skin of IMQ-treated *Camk4*−/− mice compared to IMQ-treated *Camk4*+/+ mice (Fig. 2f), reflecting that inflammation of IMQ-treated *Camk4*−/− mice are reduced compared with that in IMQ-treated *Camk4*+/+ mice.

### Fewer IL-17A+γδ TCR+ cells and DCs and more macrophages exist in the skin of IMQ-treated *Camk4*−/− mice than that of IMQ-treated WT mice

IMQ triggers psoriatic inflammation through Toll-like receptor 7[28], which is expressed on multiple myeloid-derived innate immune cells[29].

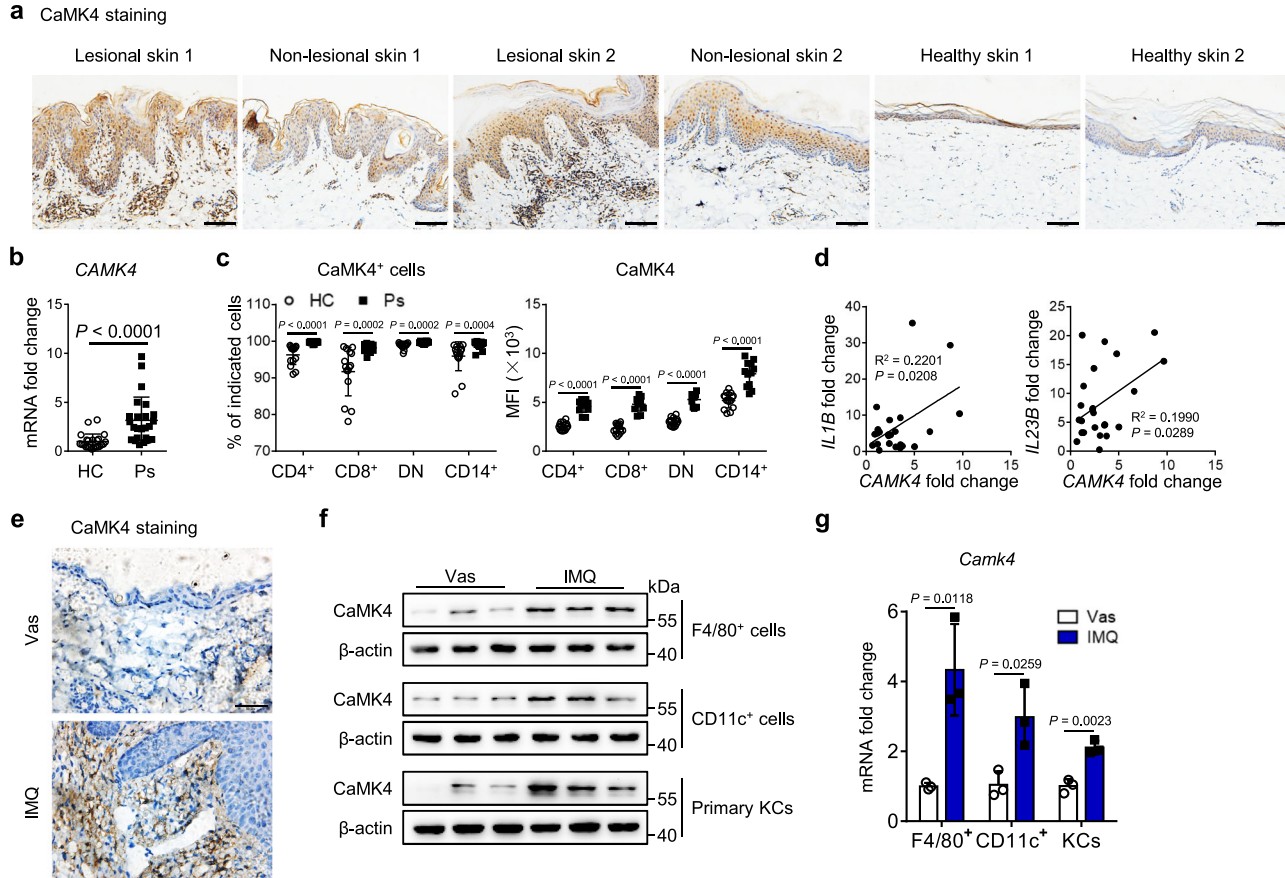

**Fig. 1 | CaMK4 expression is increased in psoriatic lesional skin from patients with psoriasis and mice with IMQ-induced psoriasis. a** Representative images of immunohistochemical staining for CaMK4 in human paired psoriatic lesional skin and non-lesional skin ($n = 4$) and healthy skin ($n = 7$). Scale bar = 100 μm. **b** The mRNA level of *CAMK4* in the cells of peripheral blood from healthy controls (HCs, $n = 24$) and patients with psoriasis (Ps, $n = 24$). **c** The proportion of CaMK4[+] cells and the MFI of CaMK4 in peripheral T cell subsets and CD14[+] monocytes from healthy controls ($n = 15$) and patients with psoriasis ($n = 12$). **d** The relationships between the expression of pro-inflammatory cytokines and *CAMK4* in the cells of peripheral blood from patients with psoriasis ($n = 24$). **e** Representative images of CaMK4-stained mouse skin sections ($n = 3$ biologically independent samples). Scale bar = 50 μm. Vas, vaseline. **f** Western blot analysis of CaMK4 expression in primary KCs and MACS-sorted F4/80[+] cells and CD11c[+] cells from mouse skin ($n = 3$ biologically independent samples). **g** Quantitative PCR analysis of *Camk4* expression as indicated ($n = 3$ biologically independent samples). Five to seven mouse skin tissues were pooled as one sample for MACS sorting, and the purity of sorted cells was >90% (**f**, **g**). Data are shown as mean ± SD. For (**b**, **c**), two-sided Mann–Whitney test; for (**d**), linear regression analysis; for (**g**), two-sided unpaired Student's *t* test. Source data are provided as a Source Data file.

We found that CaMK4 expression was upregulated in F4/80[+] and CD11c[+] cells from IMQ-treated mouse skin compared to healthy mouse skin (Fig. 1f, g). Thus, we determined whether there are associated changes in specific skin myeloid cell subsets. Representative dot plots showing flow cytometric identification of myeloid cell subsets in mouse skin were shown in Supplementary Fig. 4a. We found that the percentage and number of CD11b[+] DCs were markedly decreased, whereas the percentage and number of MHC II[+] macrophages were significantly increased in the skin of *Camk4*[−/−] mice compared to *Camk4*[+/+] mice, and there were no differences in neutrophils and F4/80[+]CD11c[+] cells between two groups of mice (Fig. 3a, b).

When innate immune cells, such as DCs, are activated; they release IL-23 and IL-1β to activate αβ T cells and γδ T cells, which produce IL-17A leading to the abnormal proliferation of KCs[4,30,31]. We then detected CD4[+] and γδ TCR[+] cell subsets in mouse skin. The gating strategy for the identification of mouse skin T cell subsets was shown in Supplementary Fig. 4b. The results showed that the percentage and number of γδ TCR[+] cells were decreased (Fig. 3c), while the percentages of IL-17A[+]CD45[+], IL-17A[+]CD4[+], and IL-17A[+]γδ TCR[+] cells were decreased in the skin of IMQ-treated *Camk4*[−/−] compared with those in IMQ-treated *Camk4*[+/+] mice (Fig. 3d). The percentages of IL-4[+]CD4[+], IFN-γ[+]CD4[+], and IFN-γ[+]γδ TCR[+] cells in the skin were not different between two groups

of mice (Fig. 3d). We also found that IL-17A in mouse skin was primarily produced by γδ TCR[+] cells, whereas IFN-γ was primarily produced by CD4[+] T cells after IMQ treatment (Fig. 3d).

Through linear regression analysis between IL-17A-producing cells and altered myeloid cells in the skin of IMQ-treated *Camk4*[+/+] and *Camk4*[−/−] mice, we noted that the percentage of CD11b[+] DCs was positively correlated with the percentages of IL-17A[+]CD4[+] ($R^2 = 0.6964$, $P < 0.0001$), IL-17A[+]γδ TCR[+] ($R^2 = 0.7193$, $P < 0.0001$), and γδ TCR[+] cells ($R^2 = 0.5512$, $P = 0.0010$) (Supplementary Fig. 5a). Nevertheless, there were negative correlations between the percentage of MHC II[+] macrophages and the percentages of IL-17A[+]CD4[+] ($R^2 = 0.2592$, $P = 0.0440$), IL-17A[+]γδ TCR[+] ($R^2 = 0.2788$, $P = 0.0355$), and γδ TCR[+] cells ($R^2 = 0.3773$, $P = 0.0144$) (Supplementary Fig. 5b). Additionally, the number of CD11b[+] DCs was positively correlated with the number of γδ TCR[+] cells ($R^2 = 0.5193$, $P = 0.0016$), whereas the number of MHC II[+] macrophages was negatively correlated with the number of γδ TCR[+] cells ($R^2 = 0.3893$, $P = 0.0098$) (Supplementary Fig. 5c).

The proportions of conventional DCs and moDCs, which drive psoriatic inflammation by producing IL-23[9,10], were reduced in the skin of IMQ-treated *Camk4*[−/−] mice compared to IMQ-treated *Camk4*[+/+] mice (Fig. 3b). Dermal macrophages have functions of scavenging and killing microorganisms and are unable to migrate and activate T cells[32].

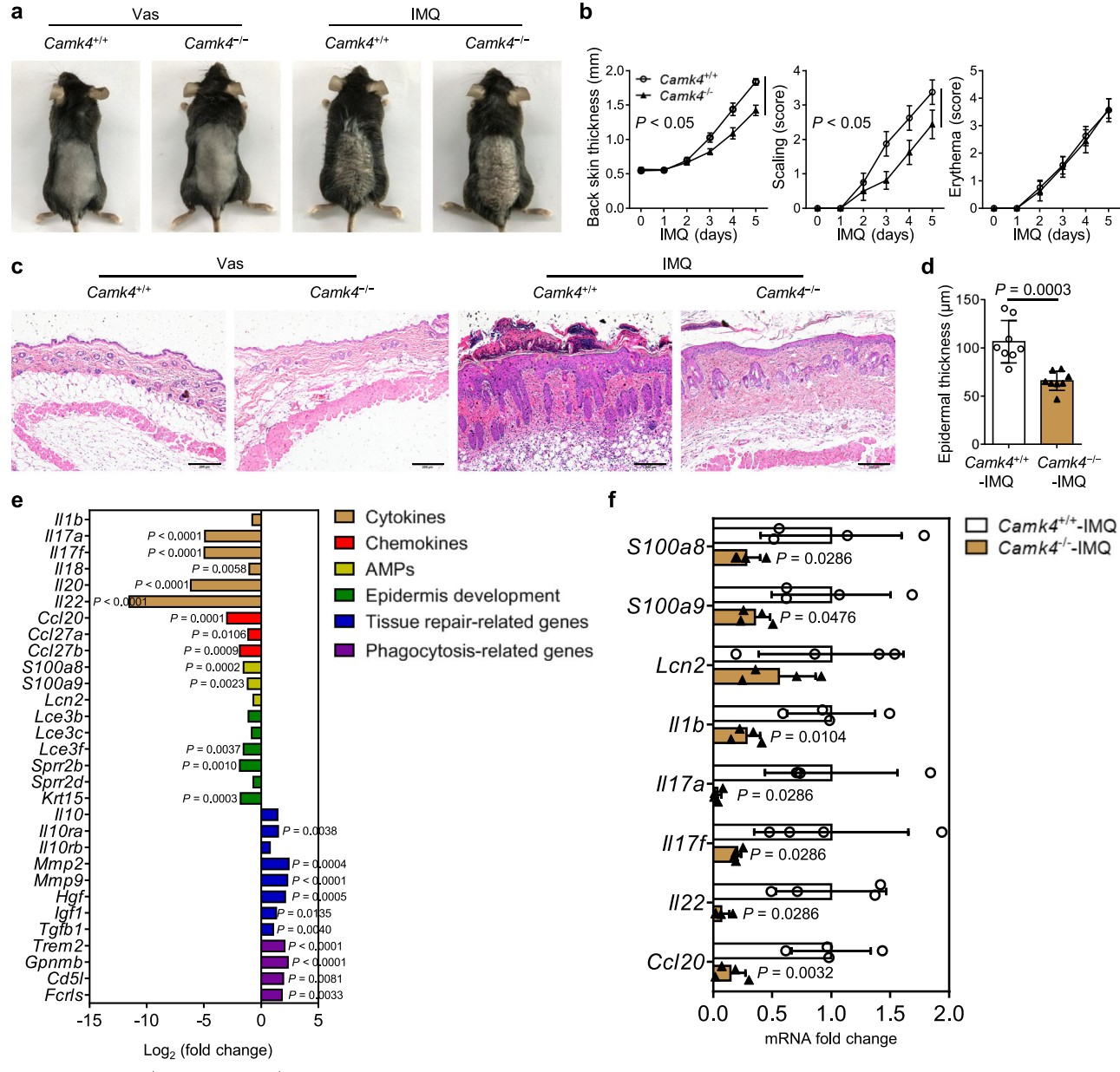

**Fig. 2 | *Camk4* deficiency alleviates the severity of IMQ-induced psoriasis.**
**a** Representative photos of mouse back skin. **b** Scoring curves of back skin thickness, scaling, and erythema. **c** H&E staining of skin sections. Scale bar = 200 μm. **d** Statistical analysis of epidermal thickness. **a**–**d** N = 7–8 per group from two independent experiments. **e** RNA sequencing analysis of the whole skin showing differentially expressed genes associated with cytokines, chemokines, AMPs, epidermis development, tissue repair, and phagocytosis between IMQ-treated *Camk4*⁺/⁺ and *Camk4*⁻/⁻ mice (n = 3 biologically independent samples). **f** The expression of pathogenic factors in the skin of IMQ-treated mice as determined by quantitative PCR (n = 4 biologically independent samples). Data are shown as mean ± SD. For (**b**, **d**, **e**), two-sided unpaired Student's *t* test; for (**f**), two-sided Mann–Whitney test and two-sided unpaired Student's *t* test. Source data are provided as a Source Data file.

In the present study, the proportions of MHC II⁺ macrophages were increased in the skin of IMQ-treated *Camk4*⁻/⁻ mice compared to IMQ-treated *Camk4*⁺/⁺ mice and exhibited a strong negative linear correlation with IL-17A⁺γδ TCR⁺ cells (Fig. 3b and Supplementary Fig. 5b). Therefore, we focused on macrophages and speculated that CaMK4 promotes psoriasis by controlling macrophages.

### CaMK4 inhibits IL-10 production by macrophages through the ADCY1-cAMP-Erk1/2 and p38 pathways to promote IMQ-induced psoriasis

Macrophages perform regulatory functions mainly through the production of IL-10[33]. This immunomodulatory cytokine is a central anti-inflammatory mediator that reduces tissue damage caused by excessive and uncontrolled inflammatory stimulation and protects the host from an excessive inflammatory response[34,35]. Thus, we analyzed IL-10 expression in the skin and in macrophages from IMQ-treated *Camk4*⁻/⁻ and *Camk4*⁺/⁺ mice, respectively. The IL-10 gene and protein levels in the skin of IMQ-treated *Camk4*⁻/⁻ mice were significantly greater than those in IMQ-treated *Camk4*⁺/⁺ mice (Fig. 4a, b). We then performed dual immunofluorescence and intracellular staining of flow cytometry experiments to explore the source of IL-10. We found that the number of F4/80⁺IL-10⁺ cells in the dermis of IMQ-treated *Camk4*⁻/⁻ mice was increased compared to IMQ-treated *Camk4*⁺/⁺ mice (Fig. 4c), indicating that the number of dermal IL-10-

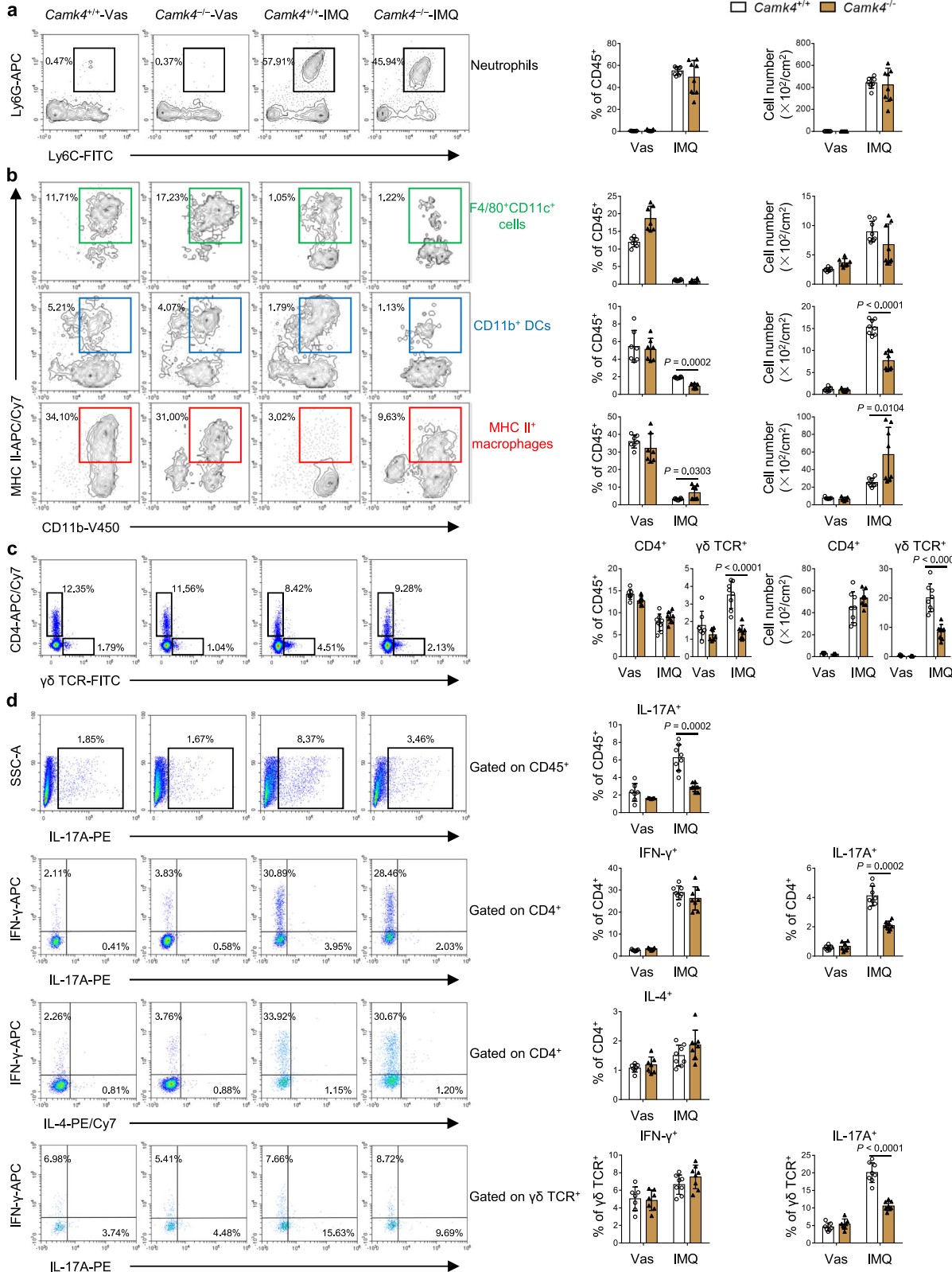

**Fig. 3 | Analysis of mouse skin myeloid cell and T-cell subsets. a** Representative flow cytometry plots and statistical analysis of neutrophils. **b** Representative flow cytometry plots and statistical analysis of F4/80+CD11c+ cells, CD11b+ DCs, and MHC II+ macrophages. **c** Representative flow cytometry plots and statistical analysis of CD4+ and γδ TCR+ cells. **a**–**c** Representative percentages indicate each subset as a proportion of total CD45+ cells. **d** Representative flow cytometry plots and statistical analysis of IL-17A+CD45+, IFN-γ+CD4+, IL-4+CD4+, IL-17A+CD4+, IFN-γ+γδ TCR+, and IL-17A+γδ TCR+ cells. **a**–**d** *N* = 7–8 per group from two independent experiments. Data are shown as mean ± SD. For (**a**–**d**), two-sided Mann–Whitney test and two-sided unpaired Student's *t* test. Source data are provided as a Source Data file.

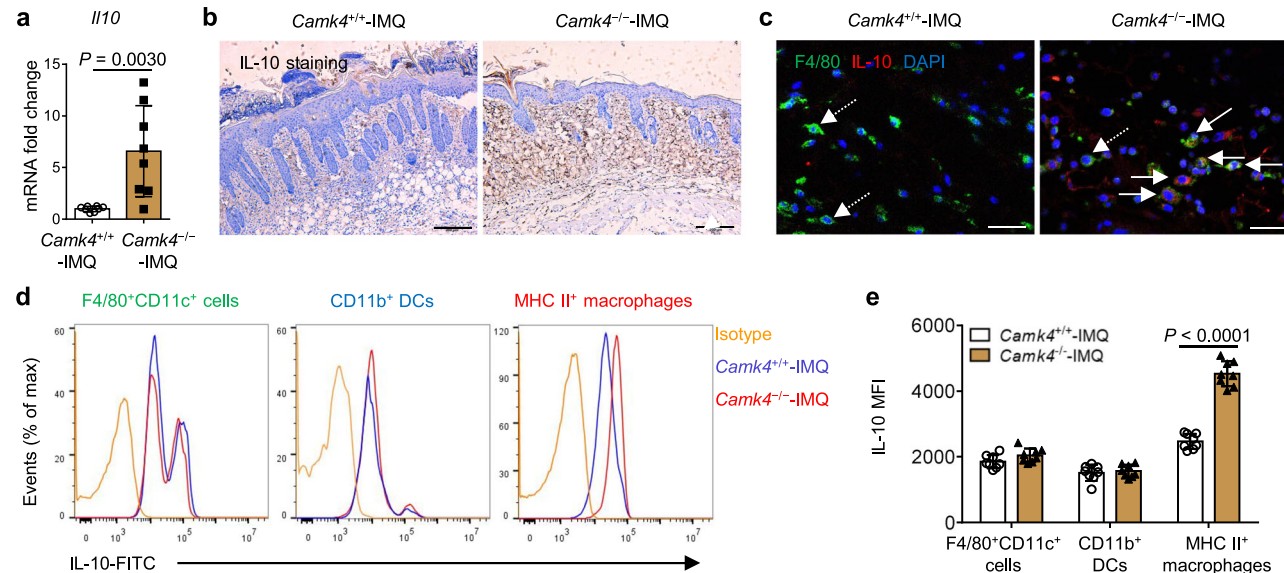

**Fig. 4 | Dermal MHC II⁺ macrophages are the main source of IL-10 in IMQ-treated *Camk4⁻/⁻* mice. a** Quantitative PCR analysis of *Il10* expression in mouse skin. **b** Mouse skin sections were stained for IL-10. Scale bar = 200 μm. **c** Immunofluorescent microscopy images of mouse skin sections stained with anti-F4/80 (green), anti-IL-10 (red), and DAPI (blue). The solid arrows show F4/80⁺IL-10⁺ cells, and the dashed arrows show F4/80⁺IL-10⁻ cells in the dermis. Scale bar = 50 μm. **d** The expression of IL-10 in F4/80⁺CD11c⁺ cells, CD11b⁺ DCs, and MHC II⁺ macrophages as determined by flow cytometry. **e** Statistical analysis of the MFI of IL-10. **a**–**e** $N = 8$ per group from two independent experiments. Data are shown as mean ± SD. For (**a**), two-sided Mann–Whitney test; for (**e**), two-sided unpaired Student's *t* test. Source data are provided as a Source Data file.

producing macrophages was increased in IMQ-treated *Camk4⁻/⁻* mice. Intracellular staining of IL-10 also showed that the amount of IL-10 in MHC II⁺ macrophages from the skin of IMQ-treated *Camk4⁻/⁻* mice was greater than that in IMQ-treated *Camk4⁺/⁺* mice (Fig. 4d, e). However, the amount of F4/80⁺CD11c⁺ cell- and CD11b⁺ DC-derived IL-10 in the skin was not different between the two groups of mice (Fig. 4d, e). In addition, we found that dermal MHC II⁺ macrophages were the main source of IL-10 in IMQ-treated *Camk4⁻/⁻* mice (Fig. 4d, e). These data reflect that CaMK4 may regulate IL-10 production in MHC II⁺ macrophages.

To investigate how CaMK4 regulates IL-10 production in macrophages, we performed a co-immunoprecipitation (co-IP) assay with CaMK4 antibody followed by LC-MS/MS using RAW264.7 cells. A partial list of proteins binding to CaMK4 was shown in Supplementary Table 1. The proteins AKT, CREM, and Calm all have been reported to interact with CaMK4. In the present study, adenylyl cyclase 1 (ADCY1) was discovered to interact with CaMK4. It has been reported that the ADCY1-cyclic adenosine monophosphate (cAMP) signaling is a positive regulator of Erk1/2 and p38[36,37], and the Erk1/2 and p38 MAPK pathways are important in *Il10* gene regulation in myeloid cells[34,35,38]. Thus, we focused on ADCY1 and explored the mechanism by which CaMK4 affects the ADCY1-cAMP-Erk1/2 and p38 pathways. We subsequently performed co-IP and western blot assays in mouse bone marrow-derived macrophages (BMDMs) to test the interaction between CaMK4 and ADCY1. As shown in Fig. 5a, CaMK4 directly interacted with ADCY1. In the subsequent CaMK4 inhibition experiments, we treated BMDMs with the CaMK4 inhibitor KN-93. KN-93 inhibits not only CaMK4 but also CaMK2; thus, we detected the expression of *Camk2a*, *Camk2b*, *Camk2d*, and *Camk2g* in BMDMs with or without IMQ treatment. Our results showed that the expression of *Camk2b* was upregulated in IMQ-treated BMDMs compared to DMSO-treated BMDMs, whereas the expression of *Camk2a*, *Camk2d*, and *Camk2g* was not different between the two groups (Supplementary Fig. 6a). Additionally, the expression of CaMK4 was upregulated in IMQ-treated BMDMs compared to DMSO-treated BMDMs (Fig. 5b, lane 2 versus lane 1). These results indicate that CaMK4 is preferentially induced and indirectly prove that KN-93 inhibits CaMK4 in IMQ-treated BMDMs. To test the

inhibition efficiency of KN-93, BMDMs were treated with KN-93. The results showed that CaMK4 protein level was decreased with the increase of KN-93 concentration (Supplementary Fig. 6b). In IMQ-treated BMDMs, KN-93 treatment markedly increased Erk1/2 and p38 phosphorylation (Fig. 5b, lane 3 versus lane 2) as well as IL-10 gene and protein levels (Fig. 5c, d, column 3 versus column 2); in addition, inhibitions of Erk1/2 and p38 blocked the increase of IL-10 gene and protein levels (Fig. 5c, d, columns 7 and 10 versus column 3). Interestingly, we found that ADCY1 expression was increased after CaMK4 inhibition (Fig. 5b, lane 3 versus lane 2). Cotreatment of the ADCY1 inhibitor ST034307 and KN-93 reduced Erk1/2 and p38 phosphorylation (Fig. 5b, lane 4 versus lane 3) as well as IL-10 gene and protein levels (Fig. 5c, d, column 4 versus column 3) compared to KN-93 treatment; meanwhile, the addition of cAMP elevated and restored Erk1/2 and p38 phosphorylation (Fig. 5b, lane 5 versus lane 3, lane 6 versus lane 4) as well as IL-10 gene and protein levels (Fig. 5c, d, column 5 versus column 3, column 6 versus column 4) compared to KN-93 treatment and cotreatment of ST034307 and KN-93, respectively; additionally, inhibitions of Erk1/2 and p38 reduced IL-10 gene and protein levels in the presence of cAMP (Fig. 5c, d, columns 8 and 11 versus column 5, columns 9 and 12 versus column 6). Taken together, these data indicate that CaMK4 downregulates the ADCY1-cAMP-Erk1/2 and p38 pathways, thus inhibiting IL-10 production in macrophages.

We further assessed the regulation of CaMK4 in macrophage polarization in vitro and found that *Camk4* deficiency had no effect on M1 macrophage polarization, but resulted in increased expression of M2 macrophage genes, including *Retnla* (Fizz1), *Chil3* (Ym1), *Tgfb1*, and *Il10* (Fig. 5e). Moreover, the expression of phagocytosis-related genes *Trem2*, *Gpnmb*, *Cd81*, *CdSl*, *Fcrls*, and *Macro* was upregulated in IMQ-treated *Camk4⁻/⁻* BMDMs compared to *Camk4⁺/⁺* BMDMs (Fig. 5f). These data demonstrate that CaMK4 restricts M2 macrophage polarization and macrophage phagocytosis.

To confirm the mechanism by which CaMK4 inhibits IL-10 production through the Erk1/2 and p38 pathways found in mouse macrophages, we sorted peripheral CD14⁺ monocytes from patients with psoriasis. Peripheral monocytes treated with KN-93 showed increased Erk1/2 and p38 phosphorylation as well as IL-10 gene and protein levels

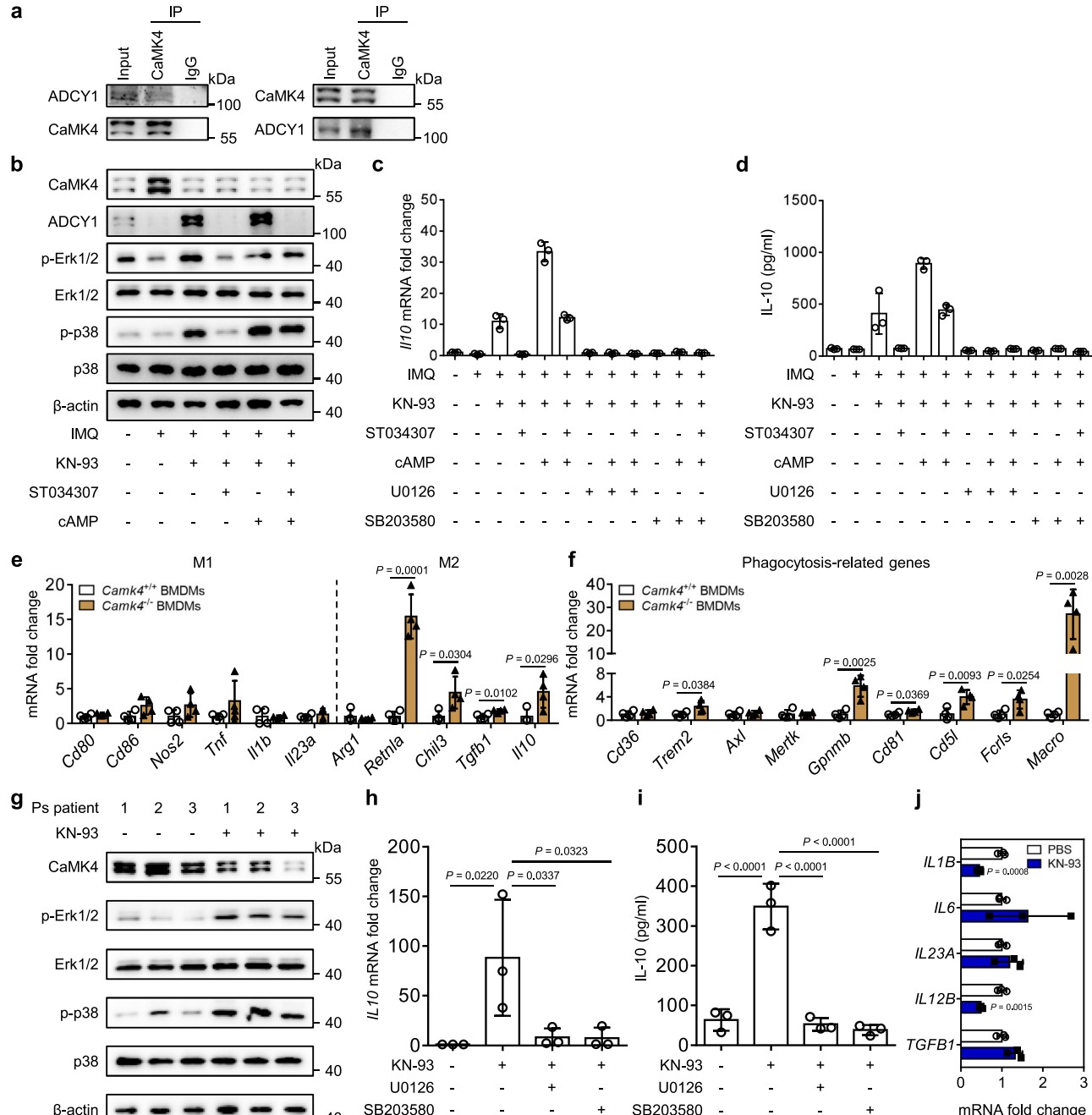

**Fig. 5 | CaMK4 inhibits IL-10 production by macrophages through the ADCY1-cAMP-Erk1/2 and p38 pathways. a** Co-IP assay of CaMK4 and ADCY1 in BMDMs. **b** Western blot analysis of CaMK4, ADCY1, p-Erk1/2, Erk1/2, p-p38, and p38 in BMDMs treated with IMQ, KN-93, ST034307, or cAMP. **c** Quantitative PCR analysis of *Il10* expression as indicated (*n* = 3 biologically independent samples). **d** ELISA analysis of IL-10 in the supernatants of BMDMs as indicated (*n* = 3 biologically independent samples). **e** Quantitative PCR analysis of macrophage phenotype markers (M1 and M2) in *Camk4*[+/+] and *Camk4*[−/−] BMDMs stimulated with IMQ (*n* = 4 biologically independent samples). **f** Quantitative PCR analysis of phagocytosis-related genes (*n* = 4 biologically independent samples). **g** Western blot analysis of

CaMK4, p-Erk1/2, Erk1/2, p-p38, and p38 in MACS-sorted peripheral CD14[+] monocytes from patients with psoriasis. **h** Quantitative PCR analysis of *IL10* in monocytes (*n* = 3 biologically independent samples). **i** ELISA analysis of IL-10 in monocyte supernatants (*n* = 3 biologically independent samples). **j** Quantitative PCR analysis of pro-inflammatory genes in monocytes (*n* = 3 biologically independent samples). The experiments in **a**–**f** were repeated three times with similar results. Data are shown as mean ± SD. For (**e**, **f**, **j**), two-sided unpaired Student's *t* test; for (**h**, **i**), one-way ANOVA with Bonferroni's post-test. Source data are provided as a Source Data file.

compared to untreated peripheral monocytes (Fig. 5g–i). Moreover, the addition of Erk1/2 and p38 inhibitors reduced IL-10 gene and protein levels (Fig. 5h, i). Furthermore, the expression of *IL1B* and *IL12B* was downregulated in KN-93-treated peripheral monocytes compared to untreated peripheral monocytes (Fig. 5j).

We then performed BMDM transfer, exogenous IL-10 addition, and endogenous IL-10 neutralization experiments in vivo. As shown in

Fig. 6, IMQ-treated WT mice that received *Camk4*[−/−] BMDMs showed less severe psoriatic inflammation and decreased mRNA levels of skin pathogenic factors, including *S100a8*, *S100a9*, *Tnf*, *Il17a*, *Il23a*, *Ccl2*, and *Ccl20*, compared with those of IMQ-treated WT mice that received *Camk4*[+/+] BMDMs. Encouragingly, *Il10* expression was significantly increased in the skin of IMQ-treated WT mice after *Camk4*[−/−] BMDM transfer (Fig. 6e). In addition, IMQ-treated WT mice that received IL-10

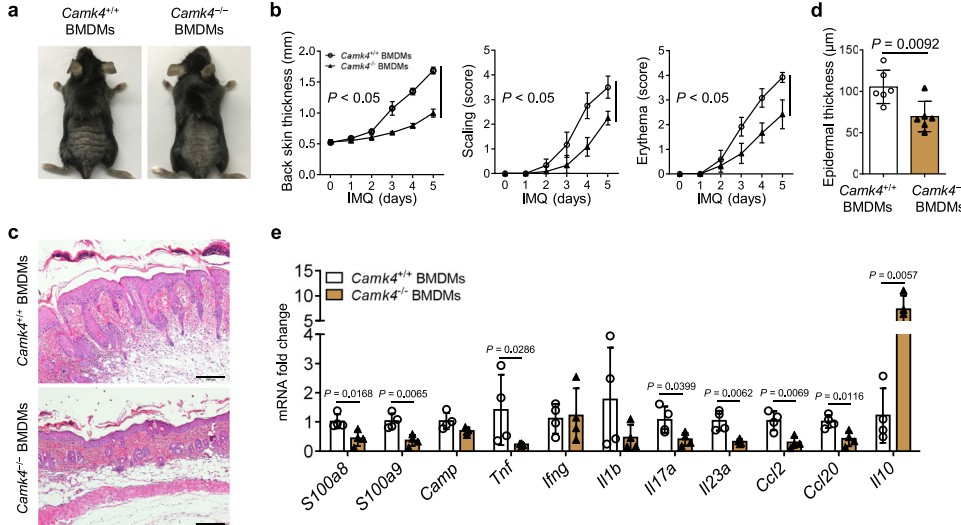

**Fig. 6 | *Camk4⁻/⁻* BMDM transfer alleviates IMQ-induced psoriatic inflammation.** A total of $1 \times 10^6$ BMDMs from *Camk4⁺/⁺* or *Camk4⁻/⁻* mice were intravenously injected into each WT mouse 1 h before IMQ treatment at day 0, and mice were euthanized after continuous application of IMQ for 5 days. **a** Representative photo of mouse back skin. **b** Scoring curves of back skin thickness, scaling, and erythema. **c** Skin histology. Scale bar = 200 μm. **d** Statistical analysis of epidermal thickness. **a**–**d** N = 6 per group from two independent experiments. **e** The expression of pathogenic factors in the skin of IMQ-treated mice as determined by quantitative PCR (n = 4 biologically independent samples). Data are shown as mean ± SD. For (**b**, **d**), two-sided unpaired Student's t test; for (**e**), two-sided Mann–Whitney test and two-sided unpaired Student's t test. Source data are provided as a Source Data file.

showed less severe psoriatic inflammation and reduced mRNA levels of skin pathogenic factors, including *Camp*, *Tnf*, *Il1b*, *Il17a*, *Il23a*, and *Ccl2*, compared to control mice receiving PBS (Supplementary Fig. 7). In contrast, IMQ-treated *Camk4⁻/⁻* mice with anti-IL-10 neutralization showed more severe psoriatic inflammation compared to control mice treated with IgG (Supplementary Fig. 8). Collectively, these results provide evidence that loss of *Camk4* restores IL-10 production in macrophages, thereby alleviating IMQ-induced psoriatic inflammation.

To directly assess the cell type-intrinsic function of *Camk4* in macrophages, we established a conditional *Camk4* knockout mouse model, named *Camk4*ᶠˡ/ᶠˡ *Lyz2*-Cre mice. These mice and control mice were treated with IMQ, and skin inflammation and gene expression were analyzed. As shown in Fig. 7a–d, loss of *Camk4* in macrophages alleviated IMQ-induced psoriatic inflammation. Accordingly, genes associated with psoriasis, including *S100a8*, *S100a9*, *Camp*, *Tnf*, *Il1b*, *Il23a*, and *Ccl2*, were strongly downregulated (Fig. 7e). The alleviation of disease was correlated with more MHC II⁺ macrophages in the skin (Fig. 7f, g).

### The CaMK4-AKT-NF-κB pathway promotes KC pro-inflammatory phenotypes

CaMK4 expression was upregulated in not only immune cells but also primary KCs in the skin of IMQ-treated mice compared to healthy mouse skin (Fig. 1f, g), which drove us to determine the mechanism of CaMK4 in KCs. To knockdown CaMK4 in the human immortalized KC cell line HaCaT, we transfected HaCaT cells with *CAMK4*-specific siRNA and then stimulated the cells with TNF and IL-17A, which are two cytokines that can mimic the internal inflammatory environment of psoriasis[21]. After TNF and IL-17A stimulation, HaCaT cells transfected with si*CAMK4* showed increased apoptosis at the early stage and late stage (Fig. 8a, b), decreased cell proliferation (Fig. 8c), and reduced expression of pro-inflammatory genes *S100A8*, *CAMP*, *DEFB4A*, *SPRR2A*, *SPRR2B*, *TNF*, *CCL2*, and *CCL20* (Fig. 8d) compared to scrambled control-transfected cells. To link apoptosis, cell cycle, and pro-inflammatory gene expression together, we first analyzed the NF-κB pathway because NF-κB induces a variety of anti-apoptotic factors to protect against TNF-induced apoptosis, regulates cyclin-dependent

kinase and cyclin family members to promote cell proliferation, and induces the transcription of multiple pro-inflammatory genes[39–41]. Also, CaMK4 has been reported to modulate the AKT pathway[24,42]. Therefore, we speculated that CaMK4 influences KC phenotypes by regulating the AKT-NF-κB pathway. The co-IP results displayed a physical association between CaMK4 and AKT (Fig. 8e). In addition, the phosphorylation levels of AKT, IKKα/β, and p65 were decreased after *CAMK4* knockdown (Fig. 8f). Taken together, these results demonstrate that CaMK4 inhibits apoptosis as well as promotes cell proliferation and the expression of pro-inflammatory genes through the AKT-NF-κB pathway in KCs.

### CaMK4 inhibitor ameliorates IMQ-induced psoriasis

To explore interventions targeting CaMK4 for psoriasis treatment, the in vivo therapeutic efficacy of the CaMK4 inhibitor KN-93 was assessed (Fig. 9a). IMQ-treated WT mice treated with KN-93 displayed less severe psoriatic inflammation and reduced mRNA levels of skin pathogenic factors, including *S100a8*, *S100a9*, *Tnf*, *Il1b*, *Il17a*, *Il23a*, and *Ccl2*, compared to IMQ-treated WT mice that received PBS (Fig. 9b–f). In addition, *Il10* expression was increased in the skin of IMQ-treated mice treated with KN-93 (Fig. 9f). In order to eliminate the influence of CaMK2, we detected the expression of *Camk2a*, *Camk2b*, *Camk2d*, and *Camk2g* in the skin of IMQ-treated mice and control mice. The expression of *Camk2a* and *Camk2b* was decreased in the skin of IMQ-treated mice compared to control mice, whereas the expression of *Camk2d* and *Camk2g* was not different between the two groups of mice (Supplementary Fig. 9a). Additionally, compared to IMQ-treated *Camk4⁻/⁻* mice, KN-93 and IMQ-treated *Camk4⁻/⁻* mice had no difference in the severity of IMQ-induced psoriasis (Supplementary Fig. 9b–e). Briefly, these data suggest that the reduction of the severity of IMQ-induced psoriasis by KN-93 is mainly caused by the inhibition of CaMK4 rather than CaMK2.

### Discussion

The contribution of CaMK4 in SLE and EAE is gradually becoming understood. CaMK4 promotes the differentiation of Th17 cells and the production of IL-17, and this cytokine is a key factor in IL-17-associated autoimmune diseases, such as SLE, EAE, and psoriasis[5,6,24,26]. The αβ

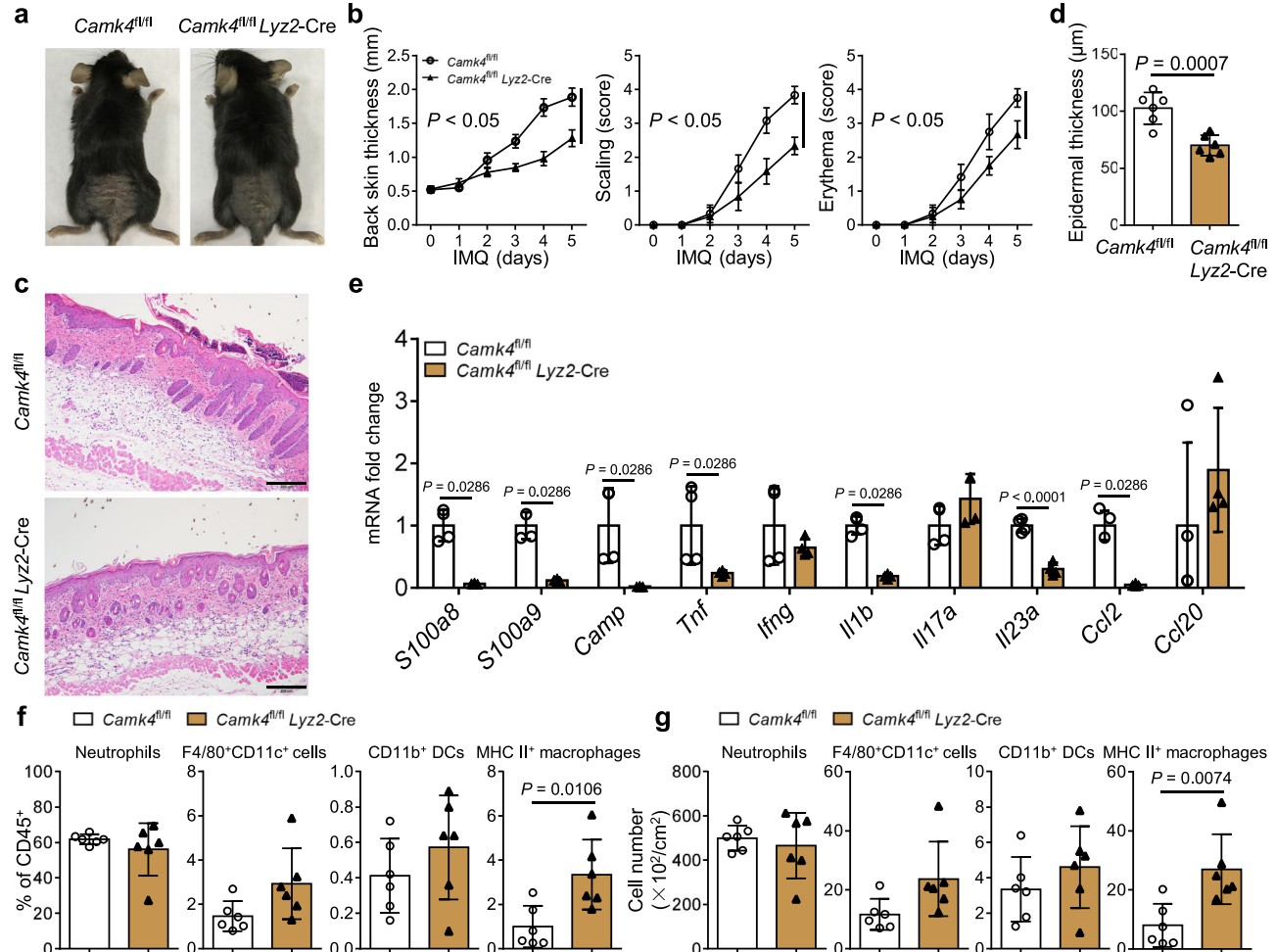

**Fig. 7 | Loss of *Camk4* in macrophages alleviates IMQ-induced psoriatic inflammation. a** Representative photos of mouse back skin. **b** Scoring curves of back skin thickness, scaling, and erythema. **c** H&E staining of skin sections. Scale bar = 200 μm. **d** Statistical analysis of epidermal thickness. **e** Quantitative PCR analysis of pathogenic factors in the skin of IMQ-treated mice (*n* = 4 biologically independent samples). **f, g** The percentages and numbers of neutrophils, F4/80⁺CD11c⁺ cells, CD11b⁺ DCs, and MHC II⁺ macrophages in the skin. **a**–**e**, **f**, **g** *N* = 6 per group from two independent experiments. Data are shown as mean ± SD. For (**b**, **d**, **f**, **g**), two-sided unpaired Student's *t* test; for (**e**), two-sided Mann–Whitney test and two-sided unpaired Student's *t* test. Source data are provided as a Source Data file.

T cells and γδ T cells respectively account for ~50% of skin T cells, and IL-17 is mainly secreted by dermal γδ T cells[30]. In the present study, we found that CaMK4 promoted IL-17 production by Th17 cells and γδ T17 cells and that IL-17A was mainly produced by γδ T cells in the skin of IMQ-treated mice. The mechanism of CaMK4 affecting psoriasis is mediated by γδ T17 cells, which is different from the function of Th17 cells in SLE.

The expression of many genes known to be associated with psoriasis, including *Lce3f*, *Sprr2b*, *Krt15*, *S100a8*, *S100a9*, *Il17a*, *Il17f*, *Il18*, *Il20*, *Il22*, and *Ccl20*, was downregulated in the skin of IMQ-treated *Camk4⁻/⁻* mice compared with those of IMQ-treated *Camk4⁺/⁺* mice. The decrease of these pathogenic factors was accompanied by reduced skin inflammation in *Camk4⁻/⁻* mice treated with IMQ. LCE3F, SPRR2B, KRT15, S100A8, and S100A9, which are released by KCs[43,44], are recognized as antigens by immune cells, particularly pattern recognition receptor-expressing antigen-presenting cells. The activated immune cells then secrete pro-inflammatory cytokines IL-17 and IL-22 to contribute to psoriasis progression. CCL20 is the ligand of CCR6, and this chemokine receptor is responsible for recruiting leukocytes to psoriatic lesional skin and is expressed on various immune cells, including IL-17-producing T cells, monocytes, LCs, and DCs[45–47]. Our results displayed that the numbers of Th17 cells, γδ T17 cells, and DCs were significantly decreased in the skin of IMQ-treated *Camk4⁻/⁻*

mice compared to IMQ-treated *Camk4⁺/⁺* mice. Therefore, it appears that CaMK4 is required for CCL20/CCR6 chemotactic cell infiltration.

In the present study, we identified mouse skin myeloid cells based on a combination of markers. Neutrophils and monocytes were identified as Ly6G⁺Ly6C^intermediate and Ly6C^hi, respectively[48–51]. Henri et al. have used numerous markers, including CD11b, CD24, CD64, Ly6C, CCR2, and MHC II, to elegantly and clearly identify skin LCs, CD11b⁺ cDCs, Ly6C^hi moDCs, Ly6C^lo moDCs, MHC II⁺ macrophages, and MHC II⁻ macrophages[32,50,52]. Here, we used F4/80, CD11c, CD11b, and MHC II to clearly identify skin DCs and macrophages based on the expression of F4/80 on tissue macrophages and CD11c on DCs[11,53–55]. Skin CD11b⁺ DCs were identified as F4/80⁻CD11c⁺CD11b⁺MHC II⁺, and MHC II⁺ macrophages were identified as F4/80⁺CD11c⁻CD11b⁺MHC II⁺. The F4/80⁺CD11c⁺ cells included LCs, moDCs, and monocyte-derived macrophages in the inflamed skin of IMQ-treated mice. We found that there were fewer CD11b⁺ DCs in the skin of IMQ-treated *Camk4⁻/⁻* mice compared to IMQ-treated *Camk4⁺/⁺* mice, which was consistent with the downregulation of the chemokine CCL20 in the skin of IMQ-treated *Camk4⁻/⁻* mice. Interestingly, the number of MHC II⁺ macrophages was markedly increased in the skin of IMQ-treated *Camk4⁻/⁻* mice compared to IMQ-treated *Camk4⁺/⁺* mice. Our in vitro experiments demonstrated that *Camk4⁻/⁻* macrophages tended to M2

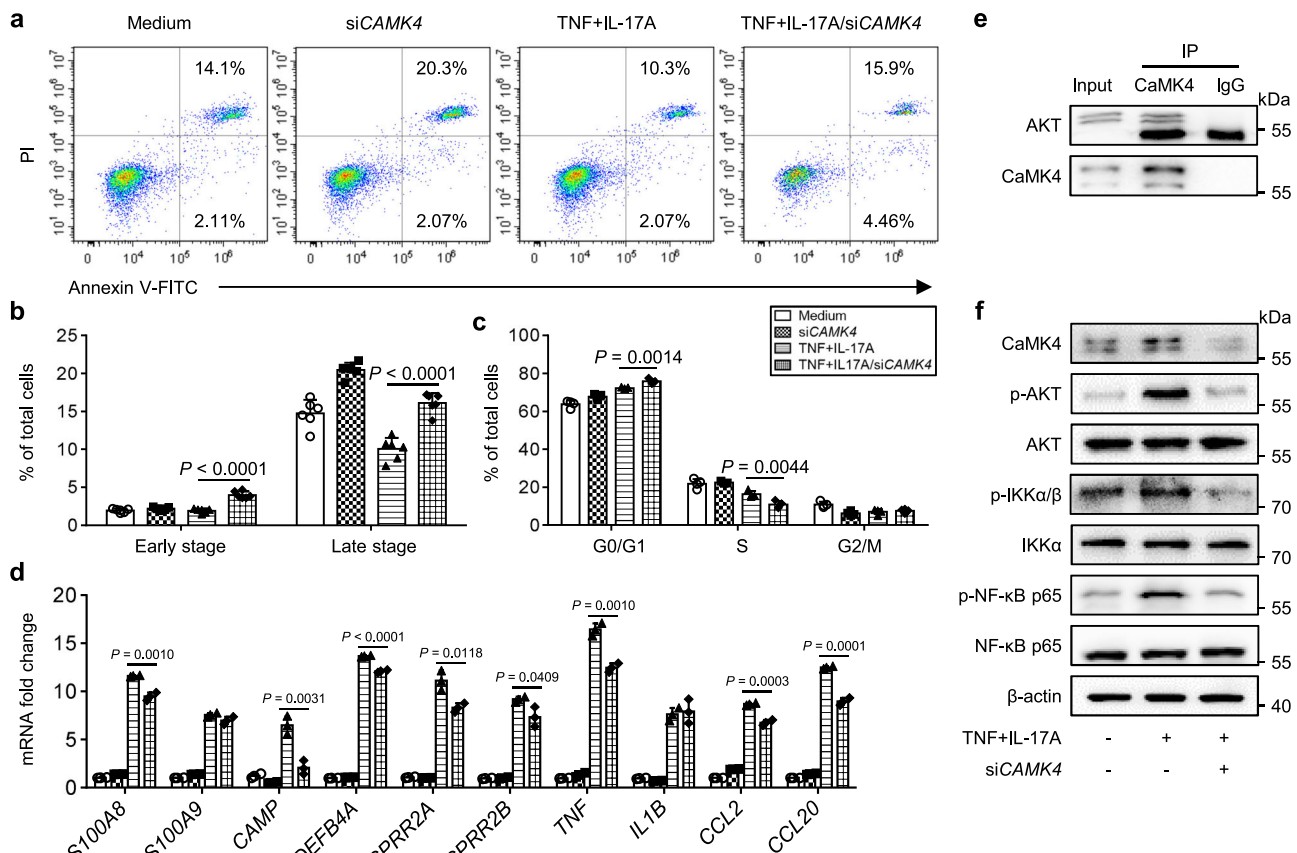

**Fig. 8 | The CaMK4-AKT-NF-κB pathway promotes KC pro-inflammatory phenotypes.** HaCaT cells were transfected with siCAMK4 or scrambled control for 24 h. Then the cells were stimulated with recombinant human TNF (50 ng/ml) and IL-17A (50 ng/ml) for 24 h. The cells were harvested for analyses of apoptosis, cell cycle, and gene and protein levels. **a** Representative flow cytometry plots of HaCaT cell apoptosis. **b** The proportions of early and late apoptotic cells ($n = 6$ biologically independent samples). **c** Statistical analysis of G0/G1, S, and M phases of cell cycle ($n = 4$ biologically independent samples). **d** The expression of pro-inflammatory genes as determined by quantitative PCR ($n = 3$ biologically independent samples). **e** Co-IP assay of CaMK4 and AKT. **f** Western blot analysis of CaMK4, p-AKT, AKT, p-IKKα/β, IKKα, p-NF-κB p65, and NF-κB p65. The experiments in **a–f** were repeated three times with similar results. Data are shown as mean ± SD. For (**a–d**), two-sided unpaired Student's $t$ test. Source data are provided as a Source Data file.

macrophage polarization. M2 macrophages have an important function in alleviating psoriatic inflammation. Miki et al. have found that 4-1BBL knockout macrophages treated with LPS plus IFN-γ showed decreased M1 macrophage polarization, whereas those treated with IL-4 showed increased M2 macrophage polarization compared to WT macrophages. This skewed M1/M2 ratio toward M2 macrophages is associated with alleviating IMQ-induced psoriasis in mice[56]. Additionally, in IL-35-treated K14-VEGF-A-Tg mice, M2 macrophages have been demonstrated to be increased and accompanied by reduced inflammation[57]. Moreover, our results showed that Camk4⁻/⁻ macrophages had a stronger phagocytic capacity than Camk4⁺/⁺ macrophages. Macrophage phagocytosis contributes to pathogen and debris clearance, wound repair, tissue remodeling, and skin homeostasis maintenance. We also found that Camk4⁻/⁻ BMDM transfer and loss of Camk4 in macrophages alleviated IMQ-induced psoriatic inflammation. Macrophages in human skin may not have a single phenotype, M1 or M2, but rather they have great phenotypic plasticity in responding to the microenvironment. Furthermore, in human psoriatic lesional skin, a subpopulation of CD163-positive macrophages is classically activated and produces inflammatory molecules[58]. However, the CD163 marker is more assimilated to "alternative" macrophages or M2 macrophages in normal human skin. There are only limited reports of macrophages in human psoriasis, possibly because there are too few macrophages and it is difficult to isolate from human skin. Single-cell sequencing of human skin may be able to elucidate the function of macrophages.

IL-10 is an anti-inflammatory cytokine that inhibits the production of pro-inflammatory cytokines and chemokines, including IL-1, IL-6, TNF, CCL2, and CCL5[59–61]. On the contrary, Hedrich et al. have reported that STAT3 promotes IL-10 expression in lupus T cells, and increased IL-10 in the serum and tissue is correlated with disease activity and tissue damage[62,63]. The opposite functions of IL-10 may be associated with the different types of cells, tissues, and diseases. The diminished level of IL-10 in the serum and skin of patients with psoriasis is critical for the induction of disease flare-ups[64,65]. IL-10 is a negative regulator of psoriasis. In the present study, we observed increased expression of IL-10 in the skin of IMQ-treated Camk4⁻/⁻ mice compared to IMQ-treated Camk4⁺/⁺ mice. Our immunofluorescent and flow cytometric analyses indicated that macrophages were the major population of IL-10-releasing myeloid cells in the skin of IMQ-treated Camk4⁻/⁻ mice. Our in vitro data confirmed that CaMK4 directly interacted with ADCY1 and downregulated the ADCY1-cAMP-Erk1/2 and p38 pathways, thus restraining IL-10 production in macrophages. However, the mechanism by which CaMK4 affects ADCY1 needs further study, and we speculate that CaMK4 may degrade ADCY1 by recruiting and phosphorylating ubiquitinase. We also found that exogenous IL-10 addition alleviates the severity of IMQ-induced psoriatic inflammation. These data suggest that after IMQ treatment, CaMK4 controls the expression of pro-inflammatory factors to promote psoriasis progression by inhibiting IL-10 production in macrophages. Notably, clinical trials of IL-10 administration for

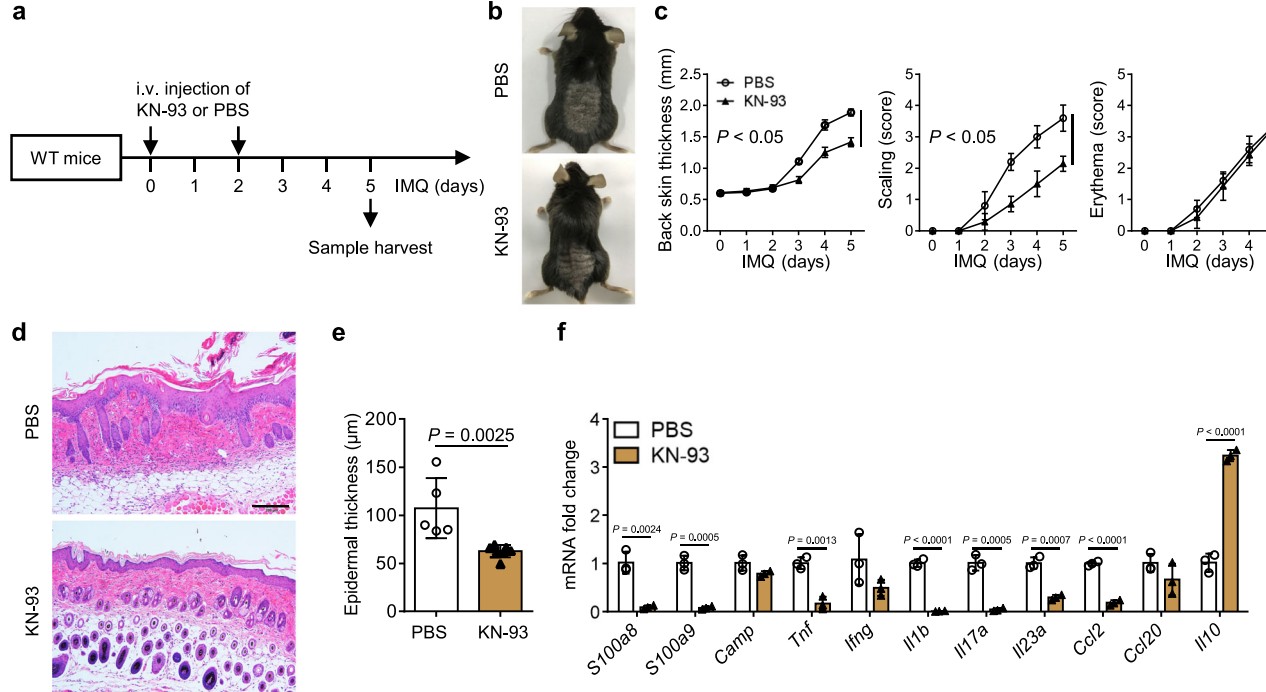

**Fig. 9 | CaMK4 inhibitor ameliorates IMQ-induced psoriasis. a** Schematic representation of the mouse model. WT mice were topically treated with 62.5 mg IMQ on shaved back skin daily for 5 consecutive days. KN-93 (0.24 mg per mouse) or PBS was intravenously injected into the mice 1 h before IMQ treatment on day 0 and day 2. Samples were harvested on day 5 for subsequent experiments. **b** Representative photos of mouse back skin. **c** Scoring curves of back skin thickness, scaling, and erythema. **d** H&E staining of skin sections. Scale bar = 200 µm.

**e** Statistical analysis of epidermal thickness. **a**–**e** $N = 5$–7 per group from two independent experiments. **f** The expression of pathogenic factors in the skin of IMQ-treated mice as determined by quantitative PCR ($n = 3$ biologically independent samples). Data are shown as mean ± SD. For (**c**), two-sided unpaired Student's $t$ test; for (**e**), two-sided Mann–Whitney test; for (**f**), two-sided Mann–Whitney test and two-sided unpaired Student's $t$ test. Source data are provided as a Source Data file.

psoriasis treatment have shown limited success, but it yields some promising results. Further studies are needed to understand the regulation of IL-10 expression and function in different contexts with the aim of developing treatments for autoimmune diseases.

We confirmed our findings in the skin and peripheral blood of patients with psoriasis. CaMK4 was highly expressed in not only psoriatic lesional skin but also peripheral immune cells, particularly monocytes. *CAMK4* expression in the cells of peripheral blood from patients with psoriasis was positively correlated with the expression of *IL1B* and *IL12B*. IL-1β and IL-12 are mainly secreted by monocytes/macrophages[66,67], and CaMK4 inhibition downregulated the expression of *IL1B* and *IL12B* in monocytes, indicating that CaMK4 is likely involved in monocyte-derived IL-1β and IL-12 production. In addition, our results showed that CaMK4, through downregulating Erk1/2 and p38 phosphorylation, inhibited IL-10 production in monocytes. Collectively, our data show that CaMK4 has critical functions in monocytes, including promoting pro-inflammatory cytokine production and inhibiting anti-inflammatory cytokine production, thus affecting psoriasis progression.

In summary, our present study provides evidence of the mechanism underlying CaMK4 in the pathogenesis of psoriasis. As shown in Fig. 10, in IMQ- or AMP-activated macrophages, CaMK4 is increased and inhibits IL-10 production through the ADCY1-cAMP-Erk1/2 and p38 pathways as well as reduces the level of IL-10 in the skin, thus allowing excessive psoriatic inflammation. CaMK4 also upregulates monocyte-derived IL-1β and IL-12 expression to stimulate the release of IL-17A by γδ T cells. In turn, IL-17A causes the hyperproliferation of KCs, which produce AMPs (such as S100A8 and LL-37) and chemokines (CCL2 and CCL20) to recruit immune cells into the derma through the CaMK4-AKT-NF-κB pathway.

## Methods

### Human participants

Peripheral blood samples of patients with psoriasis and healthy controls listed in Supplementary Table 2, came from the First Affiliated Hospital of Anhui Medical University (Hefei, China). Patients with psoriasis were diagnosed with psoriasis vulgaris by two senior dermatologists. Paraffin-embedded healthy skin tissues and paired psoriatic lesional skin and non-lesional skin tissues listed in Supplementary Table 2, were from the Pathology Laboratory of Dermatology, the First Affiliated Hospital of Anhui Medical University (Hefei, China). Written informed consent was obtained from all participants.

### Mice

*Camk4*[+/+] (or WT), *Camk4*[−/−], *Camk4*[flox/flox], and *Lyz2*-Cre mice on C57BL/6 background were purchased from GemPharmatech Corporation (Nanjing, China). *Camk4*[flox/flox] mice were crossed to *Lyz2*-Cre mice to create *Camk4*[fl/fl] *Lyz2*-Cre mice. All mice were housed under dark/light cycle of 12 h, an ambient temperature of 22–25 °C, the humidity of 30–70%, and specific-pathogen-free conditions at the Laboratory Animal Center of Anhui Medical University. All mice experiments were performed with 8–10-week-old female mice on C57BL/6 background.

### Treatment of mice

For the establishment of psoriasis mouse model, mice were topically treated with 62.5 mg commercially available 5% IMQ cream (Med-shine Pharma, Chengdu, China) on shaved 2.5 cm × 2.5 cm back skin daily for 5 consecutive days. For exogenous IL-10 administration, 1 µg recombinant mouse IL-10 (BioLegend, #575804) in 200 µl PBS or 200 µl PBS control was intravenously injected into each mouse 1 h before IMQ treatment on day 0 and day 2. For endogenous IL-10 neutralization, 0.2 mg anti-mouse IL-10 (Bio X Cell, #BE0049, JES5-2A5) or IgG in

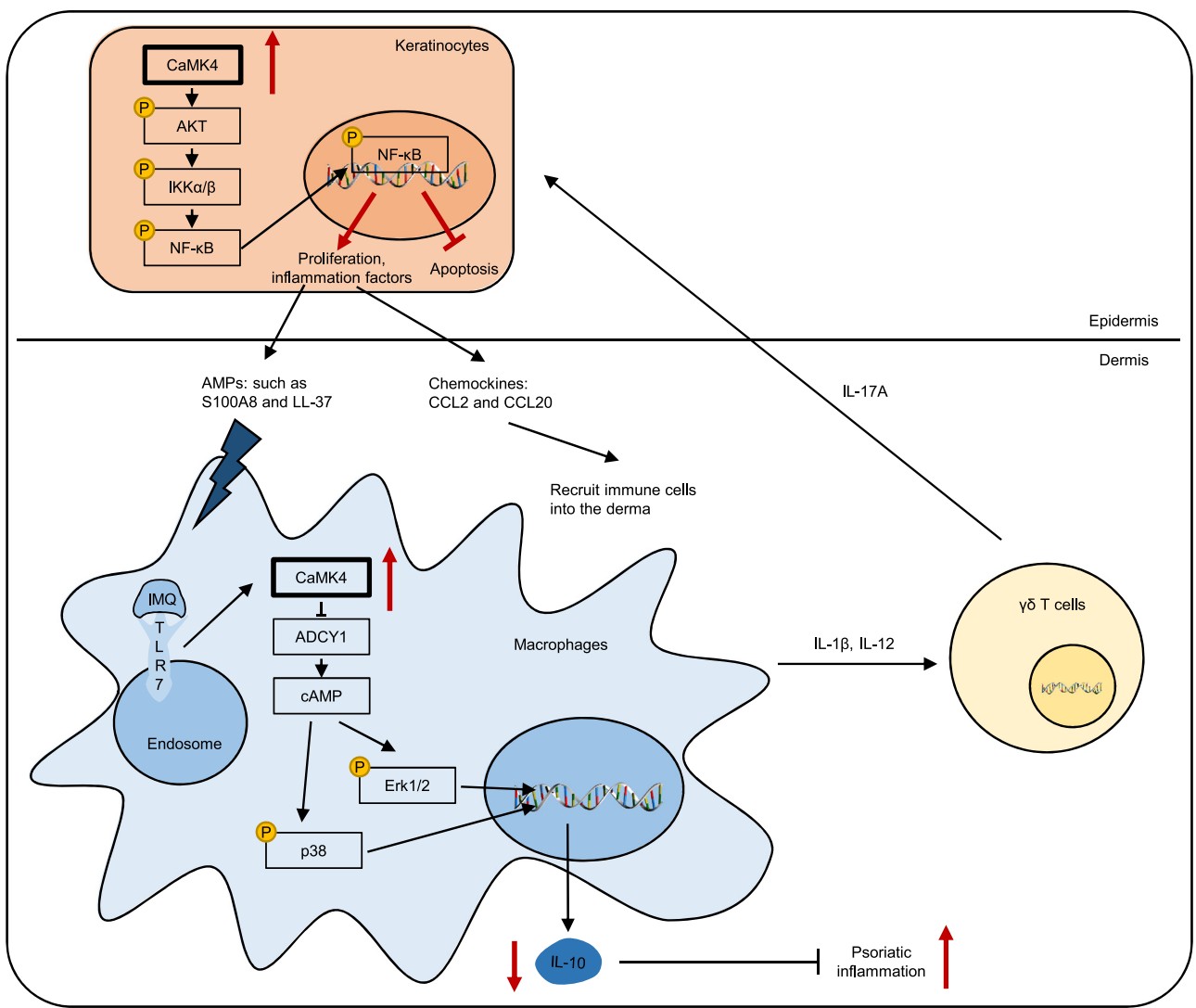

**Fig. 10 | Model diagram of the mechanism underlying CaMK4 in the pathogenesis of psoriasis.** CaMK4 is increased in IMQ- or AMP-activated macrophages and inhibits IL-10 production through the ADCY1-cAMP-Erk1/2 and p38 pathways as well as reduces the level of IL-10 in the skin, thus allowing excessive psoriatic inflammation. CaMK4 also upregulates monocyte-derived IL-1β and IL-12 expression to stimulate the release of IL-17A by γδ T cells. In turn, IL-17A causes hyperproliferation of KCs, which produce AMPs (such as S100A8 and LL-37) and chemokines (CCL2 and CCL20) to recruit immune cells into the derma through the CaMK4-AKT-NF-κB pathway.

200 μl PBS was intravenously injected into each mouse 1 h before IMQ treatment at day 0 and day 2. For inhibiting CaMK4, 0.24 mg CaMK4 inhibitor KN-93 (MedChemExpress, #HY-15465B) in 200 μl PBS or 200 μl PBS was intravenously injected into each mouse 1 h before IMQ treatment on day 0 and day 2. The severity of back skin was assessed by thickness, scaling, and erythema daily according to the clinical Psoriasis Area and Severity Index. The back skin thickness was measured using a vernier caliper. Scaling and erythema were scored from 0–4: 0, none; 1, slight; 2, moderate; 3, marked; and 4, very marked. On day 5, mice were euthanized, and samples were harvested for subsequent experiments.

**Isolation of mouse skin primary KCs and immune cells**
The isolation method of mouse skin primary KCs was described previously[68]. In brief, shaved back skin was incubated in 1% Typsin without EDTA at 37 °C for 1 h. After separating the epidermis from the dermis, the epidermis was cut into small pieces enough to enter the tip of a 10-ml pipette. Single-cell suspensions were prepared by repeated pipetting with a 10-ml pipette and filtering through a 100-μm cell strainer. Primary KCs were used for protein and RNA extractions.

For isolation of immune cells, mouse back skin was separated at the indicated time point, then cut into small pieces and incubated in RPMI 1640 medium containing 1 mg/ml collagenase IV (Sigma, #C5138), 50 μg/ml DNase I (Sangon Biotech, #A610099, Shanghai, China), 10 mM HEPES (Sangon Biotech, #E607018), and 10% FBS (Gbico, #12657-029) at 37 °C for 90 min. Digested skin pieces were passed through a 74-μm nylon mesh, and the suspensions were added additional RPMI 1640 to inactivate enzyme activity. Skin leukocytes were isolated by 30% and 70% percoll (GE Healthcare, #17-0891-01) at 1260 g for 20 min. The pellets were resuspended in PBS, and the cell number was counted.

**Flow cytometric analysis**
For mouse skin myeloid cell detection, $1 \times 10^6$ cells were blocked with purified anti-mouse CD16/32 (BD Pharmingen, #101302, 93, 1/200 dilution) and then stained with fluorochrome-conjugated monoclonal antibodies (Abs) for cell surface markers, including PerCP/Cy5.5-CD45 (BD Pharmingen, #561869, 30-F11, 1/200 dilution), V450-CD11b (BD Pharmingen, #560456, M1/70, 1/200 dilution), FITC-Ly6C (BioLegend, #128005, HK1.4, 1/200 dilution), PE-CD11c (BioLegend,

#117307, N418, 1/200 dilution), PE/Cy7-F4/80 (BioLegend, #123113, BM8, 1/200 dilution), APC-Ly6G (BioLegend, #127613, 1A8, 1/200 dilution), and APC/Cy7-MHC II (BioLegend, #107627, M5/114.15.2, 1/200 dilution). To detect myeloid cell-derived IL-10, $1 \times 10^6$ cells were blocked with purified anti-mouse CD16/32 and were stained with PerCP/Cy5.5-CD45, V450-CD11b, PE-CD11c, PE/Cy7-F4/80, APC-Ly6G, Alexa Fluor 700-Ly6C (BioLegend, #128023, HK1.4, 1/200 dilution), and APC/Cy7-MHC II, and then intracellularly stained with FITC-IL-10 (BioLegend, #505005, JES5-16E3, 1/100 dilution) after fixation and permeabilization with a Foxp3/Transcription Factor Staining Buffer Set (eBioscience, #00-5523-00).

For detecting mouse skin lymphocytes, $1 \times 10^6$ cells were stimulated with 30 ng/ml PMA (Sigma, #P8139), 1 μg/ml ionomycin (Sigma, #407951), and 5 μg/ml monensin (MedChemExpress, #HY-N0150) for 4 h. After blocking with purified anti-mouse CD16/32, the cells were surface stained with PerCP/Cy5.5-CD45, V450-CD3 (BD Pharmingen, #560804, 500A2, 1/200 dilution), APC/Cy7-CD4 (BD Pharmingen, #561830, GK1.5, 1/200 dilution), and FITC-γδ TCR (BD Pharmingen, #561996, GL3, 1/200 dilution), and then fixed and permeabilized. The cells were then incubated with APC-IFN-γ (Bio-Legend, #505809, XMG1.2, 1/100 dilution), PE/Cy7-IL-4 (BioLegend, #504117, 11B11, 1/100 dilution), and PE-IL-17A (BioLegend, #506903, TC11-18H10.1, 1/100 dilution).

For human peripheral blood sample staining, anticoagulant blood samples were lysed by ACK Lysis Buffer (0.829% NH₄Cl, 0.1% KHCO₃, 0.00372% EDTA-2Na, pH 7.2-7.4). After neutralizing and centrifugation, human peripheral leukocytes were acquired. $1 \times 10^6$ cells were blocked and then incubated with V500-CD45 (BD Pharmingen, #560777, HI30, 1/100 dilution), PerCP/Cy5.5-CD3 (BD Pharmingen, #552852, SP34-2, 1/100 dilution), FITC-CD4 (BD Pharmingen, #555346, RPA-T4, 1/100 dilution), PE/Cy7-CD8 (BD Pharmingen, #566858, HIT8α, 1/100 dilution), and PE-CD14 (BD Pharmingen, #555398, M5E2, 1/100 dilution) for cell surface markers. Then cells were fixed and permeabilized and incubated with rabbit anti-human CaMK4 (Abcam, #ab68218, EP2565AY, 1/100 dilution) and Alexa Fluor 647-conjugated goat anti-rabbit IgG H&L (Abcam, #ab150079, 1/100 dilution).

All data were acquired and analyzed using a CytoFLEX (Beckman Coulter) flow cytometer with CytExpert (version 2.4) software and FlowJo (version V10) software.

## MACS sorting and cell treatment
Five to seven mouse skin tissues were pooled as one sample for MACS sorting. Mouse skin CD3+ T cells, CD19+ B cells, F4/80+ cells, CD11c+ cells, and Ly6G+ neutrophils were sorted with PE-CD3 (BD Pharmingen, #561824, 145-2C11, 1/100 dilution), PE-CD19 (BD Pharmingen, #561736, 1D3, 1/100 dilution), PE-F4/80 (BioLegend, #123109, BM8, 1/100 dilution), PE-CD11c (BioLegend, #117307, N418, 1/100 dilution), and PE-Ly6G (BioLegend, #127607, 1A8, 1/100 dilution) using anti-PE microbeads (Miltenyi Biotech, #130-048-801) and a MACS system (Miltenyi Biotech, #130-042-201). The purity of sorted cells was >90%. Sorted mouse skin cells were used to extract protein and RNA.

After PBMC preparation, human peripheral monocytes from patients with psoriasis were sorted with PE-CD14 (BD Pharmingen, #555398, M5E2, 1/100 dilution) using anti-PE microbeads and a MACS system. The purity of sorted cells was >90%. Monocytes were treated with KN-93 (10 μM), MEK1/2 (MEK1/2 is upstream of Erk1/2) inhibitor U0126 (20 μM, Sigma, #U120), or p38 inhibitor SB203580 (20 μM, Sigma, #S8307) to extract protein and RNA for 12 h or to test cytokines in the supernatants for 24 h.

## BMDM culture and treatment
Bone marrow cells were flushed from tibias and femurs of *Camk4*+/+ and *Camk4*-/- mice and resuspended in RPMI 1640 medium supplemented with 10% FBS, 1% antibiotics, and 10 ng/ml M-CSF (Peprotech,

#315-02) at 37 °C with 5% CO₂. The culture medium was changed every 2 days for 6 days. BMDMs were identified as F4/80 and CD11b double-positive. *Camk4*+/+ BMDMs were treated with KN-93 (10 μM), ADCY1 inhibitor ST034307 (20 μM, MedChemExpress, #HY-101279), cAMP (2 μg/ml, Macklin, #A804842, Shanghai, China), U0126 (20 μM), or SB203580 (20 μM) for 1 h and followed by stimulation with IMQ (2 μg/ml, MedChemExpress, #HY-B0180A) for 12 h for protein and RNA extractions or for 24 h for measurement of cytokines. BMDMs from *Camk4*+/+ and *Camk4*-/- mice were stimulated with IMQ (2 μg/ml) for 12 h for RNA extraction.

## Adoptive transfer of BMDMs
A total of $1 \times 10^6$ BMDMs from *Camk4*+/+ or *Camk4*-/- mice were intravenously injected into each WT mouse 1 h before IMQ treatment at day 0, and mice were euthanized after continuous application of IMQ for 5 days.

## Cell lines
Human immortalized KC cell line HaCaT were cultured in DMEM supplemented with 10% FBS and 1% antibiotics. HaCaT cells were transfected with siRNA targeting *CAMK4* (si*CAMK4*, 5′-CCAUUGU-GUACAGAUGCAATT-3′, designed and synthetized by GenePharma, Shanghai, China) or scrambled control (5′-UUCUCCGAACGUGU-CACGUTT-3′) for 24 h. Then the cells were stimulated with recombinant human TNF (50 ng/ml, Peprotech, #300-01A) and IL-17A (50 ng/ml, Peprotech, #200-17) for 24 h. The cells were harvested for analyses of apoptosis, cell cycle, and gene and protein levels. Cell apoptosis and cell cycle were measured by commercially available kits (cell apoptosis kit: BestBio, #BB-4101, Shanghai, China; cell cycle kit: Beyotime, #C1052, Shanghai, China) according to the manufacturer's instructions.

Mouse macrophage cell line RAW264.7 was cultured in DMEM supplemented with 10% FBS and 1% antibiotics.

## RNA sequencing
Total RNA was extracted from the whole skin of IMQ-treated *Camk4*+/+ and *Camk4*-/- mice. After quality control of RNA amount, purity, and integrity, cDNA library with 300 ± 50 bp size was generated from ~1 μg of total RNA. Then library was sequenced on an Illumina Novaseq 6000 using 2 × 150 bp paired-end sequencing chemistry. Differentially expressed genes were defined as fold change >2 or fold change <0.5 and $p < 0.05$, and then Gene Ontology (GO) and Kyoto Encyclopedia of Genes and Genomes (KEGG) pathway enrichment analyses were done. All services were provided by LC Biotech Corporation (Hangzhou, China).

## RNA isolation and quantitative PCR
Total RNA from the whole skin, cells, or whole blood was isolated with Trizol (Invitrogen, #15596018) according to the manufacturer's instructions. RNA was reverse-transcribed to cDNA using *Evo M-MLV* RT Kit with gDNA Clean for qPCR (Accurate Biology, #AG11728, Changsha, China), and quantitative PCR was performed by SYBR Green Premix *Pro Taq* HS qPCR Kit (Accurate Biology, #AG11701) with a CFX96 Real-Time System (Bio-Rad). Gene expression was calculated by Microsoft Excel (version 2013) using the $2^{-\Delta\Delta CT}$ method relative to the housekeeping gene GAPDH. All primers listed in Supplementary Tables 3 and 4 were synthesized by Sangon Biotech Corporation.

## Western blot
The cells were lysed in RIPA buffer (Beyotime, #P0013C) containing PMSF (Beyotime, #ST506) and protease and phosphatase inhibitor cocktail (Beyotime, #P1008) for protein extracts. After protein quantification, 20 μg/lane proteins were loaded and subjected to 10% SDS/PAGE gel. The proteins in the gel were transferred onto nitrocellulose membranes and then immunoblotted. The following primary and

secondary Abs were used in this study: CaMK4 (anti-mouse CaMK4: Santa Cruze, #sc-55501, H-5, 1/100 dilution; anti-human CaMK4: Abcam, #ab68218, EP2565AY, 1/100000 dilution), ADCY1 (Santa Cruze, #sc-365350, F-10, 1/100 dilution), p-Erk1/2 (Cell Signaling Technology, #4370T, D13.14.4E, 1/1000 dilution), Erk1/2 (Cell Signaling Technology, #4695T, 137F5, 1/1000 dilution), p-p38 (Cell Signaling Technology, #4511T, D3F9, 1/1000 dilution), p38 (Cell Signaling Technology, #8690T, D13E1, 1/1000 dilution), p-AKT (Cell Signaling Technology, #4058S, 193H12, 1/1000 dilution), AKT (Cell Signaling Technology, #4691S, C67E7, 1/1000 dilution), p-IKKα/β (Cell Signaling Technology, #2697S, 16A6, 1/1000 dilution), IKKα (Cell Signaling Technology, #61294S, D3W6N, 1/1000 dilution), p-NF-κB p65 (Cell Signaling Technology, #3033T, 93H1, 1/1000 dilution), NF-κB p65 (Cell Signaling Technology, #6956T, L8F6, 1/1000 dilution), β-actin (Cell Signaling Technology, #4970S, 13E5, 1/1000 dilution); HRP-conjugated goat anti-mouse IgG (H + L) (ZSGB-BIO, #ZB-2305, 1/20000 dilution, Beijing, China) and HRP-conjugated goat anti-rabbit IgG (H + L) (ZSGB-BIO, #ZB-2301, 1/20000 dilution). The signal was detected by ECL solution (Thermo Fisher, #34095), and the images were visualized using a FluorChem FC3 imaging system (ProteinSimple).

### Co-IP and LC-MS/MS

For detecting the interaction of CaMK4 and AKT in HaCaT cells, prepared cell lysates were incubated with CaMK4 Ab (Abcam, #ab68218, EP2565AY, 1/60 dilution) or rabbit IgG overnight at 4 °C with mixing and then incubated with Protein A/G Magnetic Beads (Thermo Fisher, #88802) at 4 °C for 1 h with mixing. Beads were washed three times and suspended in SDS−PAGE sample buffer. After incubating at room temperature for 10 min with mixing and magnetically separating the beads, the supernatants were collected for western blot analysis with AKT Ab.

For detecting proteins binding to CaMK4 by LC-MS/MS in RAW264.7 cells, after co-IP assay with CaMK4 Ab or rabbit IgG, the supernatants were collected for LC-MS/MS. LC-MS/MS was conducted by staff at Applied Protein Technology Corporation (Shanghai, China). The validation experiment of protein interacting with CaMK4 was performed in BMDMs, and forward and reverse co-IP assays were both done.

### ELISA

Mouse IL-10 (R&D Systems, #M1000B) and human IL-10 (Dakewe, #1111002, Shenzhen, China) in the supernatants were measured by the ELISA kits according to the manufacturer's instructions.

### Histological, immunohistochemical, and immunofluorescent assays

Skin tissues were fixed in 10% formalin, embedded in paraffin, and cut into 4-μm sections for hematoxylin and eosin (H&E) staining. The images were obtained using an upright microscope (Olympus BX53) with cellSens (version 1.5) software in a 10-fold field. The epidermal thickness was averagely calculated on three randomly selected areas from three fields per mouse by two blinded observers.

For immunohistochemistry, deparaffinized sections were boiled for 15 min in 10 mM sodium citrate buffer (pH 6.0) for antigen retrieval. Endogenous peroxidize activity was inhibited by 3% hydrogen peroxide, and nonspecific binding was blocked using a blocking solution (Beyotime, #P0260). Then sections were incubated with primary Abs (rabbit anti-human CaMK4: Abcam, #ab68218, EP2565AY, 1/400 dilution; mouse anti-mouse CaMK4: Santa Cruze, #sc-55501, H-5, 1/50 dilution; rat anti-mouse IL-10: Santa Cruze, #sc-52561, JES5-2A5, 1/50 dilution) at 4 °C overnight. HRP-conjugated goat anti-rabbit IgG (H + L) (ZSGB-BIO, #ZB-2301, 1/200 dilution), HRP-conjugated goat anti-mouse IgG (H + L) (ZSGB-BIO, #ZB-2305, 1/200 dilution), and HRP-conjugated goat

anti-rat IgG (H + L) (ZSGB-BIO, #ZB-2307, 1/200 dilution) were used for secondary Abs. Immunostaining was developed using DAB solution (ZSGB-BIO, #ZLI-9017). The images were obtained using an upright microscope (Olympus BX53) with cellSens (version 1.5) software in 10-fold, 20-fold, or 40-fold field.

For immunofluorescence, step to nonspecific binding was the same as immunohistochemistry except for inhibiting endogenous peroxidize activity. After blocking, sections were incubated with rabbit anti-mouse F4/80 (Cell Signaling Technology, #70076S, D2S9R, 1/200 dilution) and rat anti-mouse IL-10 (Santa Cruze, #sc-52561, JES5-2A5, 1/50 dilution), mouse anti-human CD68 (Abcam, #ab201973, 3F7D3, 1/80 dilution) and rabbit anti-human CaMK4 (Abcam, #ab68218, EP2565AY, 1/100 dilution) at 4 °C overnight. Secondary Abs were used including FITC-conjugated goat anti-rabbit IgG (H + L) (ZSGB-BIO, #ZF-0311, 1/100 dilution) and Rhodamine-conjugated goat anti-rat IgG (H + L) (ZSGB-BIO, #ZF-0318, 1/100 dilution), FITC-conjugated goat anti-mouse IgG (H + L) (ZSGB-BIO, #ZF-0312, 1/100 dilution) and Rhodamine-conjugated goat anti-rabbit IgG (H + L) (ZSGB-BIO, #ZF-0316, 1/100 dilution). Nuclei were stained with DAPI (Beyotime, #P0131). The images were observed and photographed using a confocal scanning microscope (Leica TCS SP8) with LAS X (version 3.5.1.18803) software in a 40-fold field.

### Statistics

All data were expressed as mean ± SD and analyzed using GraphPad Prism 6 (version 6.01) software. For comparing variables between two groups, if the data followed Gaussian distribution, a two-sided unpaired Student's $t$-test was used; if not, a two-sided Mann-Whitney test was used. One-way ANOVA with Bonferroni's post-test was used for comparisons between more than two groups. $P < 0.05$ was considered to be significant. Correlation analysis was made using linear regression analysis and the linear correlation index $R^2$ and $P$ values were calculated.

### Study approval

All animal experiments were approved by the Institutional Animal Care and Use Committee of Anhui Medical University (Approval number: LLSC20190208) and conformed to the guidelines outlined in the Guide for the Care and Use of Laboratory Animals. All efforts were made to minimize suffering. Human participant studies were approved by the Institutional Ethics Committee of Anhui Medical University (Approval number: 20190195) and were performed in accordance with the principles of the Declaration of Helsinki.

### Reporting summary

Further information on research design is available in the Nature Research Reporting Summary linked to this article.

### Data availability

The RNA sequencing data generated in this study have been deposited in NCBI Gene Expression Omnibus database under accession code GSE204832. Source data are provided in this paper.

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

## Acknowledgements

This work was supported by grants from the National Natural Science Foundation of China (81972927 and 81773313 to L.S.), Anhui Institute of Translational Medicine (ZHYX2020A005 to L.S.), the University Synergy Innovation Program of Anhui Province (GXXT-2020-064 to L.S.), and Clinical medicine discipline construction project of Anhui Medical University (2021lcxk008 to L.S.).

## Author contributions

L.S. conceived the experiments, supervised the research, and analyzed data. L.Y., Yf.Y., and B.L. designed and conducted experiments and analyzed data. L.C., R.Z., Z.L., Y.W., W.F., and D.W. fed and identified mice. H.G., Q.Z., C.Z., S.C., and W.C. coordinated clinical investigation and collected clinical samples. Y.M., Yx.Y., S.L., and Y.B. performed immunohistochemistry and immunofluorescence. M.L. and J.S. provided an excellent experimental platform and environment. L.Y., Yf.Y., B.L., and L.S. wrote the manuscript. All authors reviewed the manuscript.

## Competing interests

The authors declare no competing interests.
