## [Peer Review File · Nature Communications]

Calcium/calmodulin-dependent protein kinase IV promotes imiquimod-induced psoriatic inflammation via macrophages and keratinocytes in miceREVIEWER COMMENTS

Reviewer #1 (Remarks to the Author):

This study represents an extension of studies which were reported previously and had established the role of CaMK4 in the expression of lupus, lupus nephritis and EAE.

1. Fig1. Neutrophils are known to be important in the skin pathology in psoriasis. The authors have ignored their presence. They should be included in this figure and in subsequent sets of data.

2. Fig5. There are a number of issues. First KN93 inhibits phosphorylation of CaMK4 and not protein levels (one report, <https://bmcanesthesiol.biomedcentral.com/articles/10.1186/s12871-016-0191-4>). Therefore, something is wrong here. The data here and in figure 7 are inferential. Production of IL10 should and be studied better. A major transcription factor for IL10 is STAT3 (PMID: 27885845) which is generated in response to IL23 and there is lots of IL-23 in the psoriatic skin lesions. This reviewer would like to understand why would IL23 signal differently in different myeloid cells present in the skin

3. Fig6. Interestingly, they show that Camk4-KO and KN93-treated mice have decreased skin scaling and thickness (keratinocyte activation) BUT NOT erythema (local inflammation). However, when they treat mice with IL-10 (or inject an important quantity of Camk4-KO macrophage) there is a decrease in both skin scaling/thickness AND erythema. If IL-10 mediated the effect in Camk4-KO/KN93-treated mice, we would expect to see a decrease in erythema in these. The authors might want to confirm that it's macrophage IL-10 which mediates improvement of the disease (eg, use myeloid cell-specific IL10-KO).

4. KN93 inhibits both CaMK2 and 4. Have the authors checked the levels on CaMK2?

5. In lupus, anti-CD3/TCR antibodies (ref 27) and possible autoantigen drive the phosphorylation of CaMK4. What drives the CaMK4 in psoriasis? Someone should assume TLR7 (the receptor for imiquimod). I wonder why the authors did not pursue this pathway.

Reviewer #2 (Remarks to the Author):

This is a paper by Liang Yong et al. on the possible role of Calcium/calmodulin-dependent protein kinase IV in psoriatic inflammation. First, the authors show by immunohistochemistry CaMK4 expression in psoriasis lesional skin. They also show that CaMK4 is present in peripheral blood mononuclear cells by flow cytometry. CaMK4 seems to be higher expressed in peripheral monocytes from psoriasis patients compared to healthy controls. In the psoriasis-like imiquimod (IMQ) mouse model, the authors found high expression of CaMK4, not only in T cells but also in F4/80+ cells, CD11c+ cells and keratinocytes. When using CaMK4 knockout mice, these mice developed reduced back skin thickness and scaling compared to wildtype mice. By RNA-seq analysis and quantitative PCR a reduction of S100a8, S100a9, Il1b, Il17a, Il17f, Il22, and Ccl20 was found in IMQ-treated Camk4 knockout mice. The authors then show that CaMK4 is also expressed in F4/80 and CD11c positive cells in the IMQ model and that the frequencies of DCs are decreased in Camk4 knockout mice. Finally, the authors provide data suggesting that CaMK4 in macrophages regulates IL-10 expression and that this regulation is mediated through the CaMK4 ADCY1 cAMP Erk1/2 and p38 pathways. For this, they used pharmacological inhibitors.

Although this paper is interesting, the authors do not provide a novel mechanism, which could be of therapeutic use. First, they do not provide evidence that CaMK4 specifically

in macrophages regulates psoriatic inflammation through IL-10. For this, macrophages with genetic deletion in CaMK4, IL-10 or both would be helpful. Likewise, a transfer model could be helpful. Second, the results shown in the IMQ model may be very different from the situation in humans with psoriasis, where macrophages seem not to play a major role. Moreover, IL-10 administration was of no benefit in randomized and controlled human trials in psoriasis. If macrophages, IL-10 and cAMP are of importance in psoriasis, the authors should be able to find a regulation of these cellular and non-cellular factors in patients treated with apremilast.

The title of the paper is misleading. CaMK4 cannot mediate psoriasis inflammation.

Figs.1 and 2: The authors should provide higher resolution histology images.

Fig.3: Flow cytometry dot plots of cytokine positive CD4+ T cells and gd T cells have to be added to this figure.

Figs.3 and 4: The authors indicate that the data is from two experiments with 7 to 8 mice per group. The figures displayed show only 4 dots per group. Please show all 7 to 8 dots and describe the statistical test used in each figure legend.

Fig.5: In A the authors show beta actin as protein load, but the band is only visible in the input lane. Please explain or use a nuclear load control.

Fig.7: The Western blot analysis depicted has different backgrounds. The authors should provide the original non-cropped blots.

The statistics used in most of the figures seem not to be appropriate. For example the authors have groups of 3 or 4 data points and used a t-test, which requires Gaussian distribution.

Reviewer #3 (Remarks to the Author):

The manuscript by Yong et al explores the function of CaMK4 in the development of psoriasis. They find that Camk4-deficient animals show slightly diminished severity of imiquimod (IMQ)-induced psoriasis. They suggest this be dependent on enhanced expression of IL-10 by Camk4-deficient myeloid cells and show that administration of IL-10 or of Camk4-deficient BMDMs can diminish the severity of IMQ-induced psoriasis. They also show that treatment of animals with KN-93, an inhibitor of CaM kinases, also diminishes severity of IMQ-induced psoriasis.

The phenotype of CamK4-deficient mice in IMQ-induced psoriasis is rather mild and not entirely surprising, considering the known involvement of CamK4 in IL-17A-mediated inflammatory pathologies. Also, the suggested mechanism is not convincingly demonstrated. On the other hand, the biochemical characterization of the function of CaMK4 and IL-10 regulation in BMDMs is quite interesting.

There are a number of important concerns that need to be addressed prior publication

Main points:

1) The authors suggest that IL-10 upregulation in CaMK4-deficient animals is responsible for the reduced severity of IMQ-induced psoriasis, but do not formally prove it. In Fig. 7 they show that administration of IL-10 or injection of CamK4-deficient BMDMs diminishes the severity of psoriasis. However, in order to convincingly demonstrate that enhanced expression of IL-10 in the absence of CamK4 is responsible for the partial protection from psoriasis, they need to block IL-10 with a neutralizing antibody in CamK4-KO mice to see whether that restores normal severity to psoriasis.

2) The characterization of the immune cells that infiltrate the skin upon IMQ-treatment by flow cytometry is rather poor. The main populations found in the lesional skin of IMQ-treated mice are not quantified. The gating strategy of myeloid cells shown in Fig. S3 appears inaccurate and incomplete. Langerhans cells are gated F4/80+ CD11c+ MHCII+, but in the inflamed skin of IMQ-treated mice this gate would probably also include monocyte-derived dendritic cells and/or monocyte-derived macrophages (moDCs). A langerin staining should be included or the claim should be rephrased. Moreover, the flow cytometry analysis shown in Fig 3A and supp Fig 3. does not allow interpretation of the data and drawing the conclusions that the authors propose. If they want to highlight expansion or reduction of a certain population in the skin of CamK4-deficient animals, they should show the percentage of the indicated population among total CD45+ cells in the skin and total cell numbers (as it is back skin, probably the best way would be to show number of cells per gram of tissue). Otherwise, the values shown in Fig. 3A are meaningless and cannot be interpreted, unfortunately. Accordingly, the statistical analysis in Fig 3E is meaningless.

Infiltration of T cells (ab and gd TCR) and neutrophils have to be quantified: percentages (among CD45+ cells) and total numbers of these cells in the skin need to be shown to allow a more accurate interpretation of the phenotype of Camk4-Ko mice. Surprisingly, looking at the dot plot, the number of neutrophils appear to be very low, which is contrast to many other reports.

3) The biochemical analysis performed in BMDM (Fig. 5) shows some of the more interesting data of the article. It suggests that CamK4 upregulation upon IMQ-treatment blocks IL-10 expression by inhibition of ADCY1/cAMP production and the activation of Erk and p38 in BMDM. However, the comparison of Fig. 5C and 5E shows that treatment of BMDMs with KN-93 has a more pronounced effect on IL-10 transcript expression than genetic ablation of CamK4. This is probably due to the fact that KN-93 inhibits not only CaMK4 but also other CaM kinases (such as CaMK1 and 2, Pellicena and Schulman, Front Pharmacol 2014) which probably also mediate part of the phenotype shown here. The broad specificity of KN-93 prompts important questions that need to be addressed: i) how is the expression of other CaM kinases affected by KN-93 treatment? ii) Does KN93-treatment affect psoriasis development in IMQ treated CaMK4-/- mice? . iii) how does the deficiency of CamK4 in BMDMs affect the expression of ADCY1, p-Erk, p-p38 and IL-10 (protein) upon IMQ treatment?

4) In Figure 7, the authors aim to determine the function of CamK4 in keratinocyte and to this end they transfect HaCaT cells with CaMK4 siRNA and observe mild effects on the expression of inflammatory mediators. However, the authors do not provide evidence of the efficiency of knock-down (i.e. Western Blot analysis of CaMK4 in siRNA and scrambled treated cells). Also, they depict a cell cycle analysis in Fig. 7C, but there is no information in the manuscript on how this was done. This should be rectified.

Minor points:

5) There is an inconsistency in the gating strategy shown in Fig S1B. On the left panel, DN T cells are shown to be gated as CD45+ CD3+ CD4- CD8- cells. However, on the right panel these cells are shown as CD3-. A closer look at the axes shows that the scale for CD3 in the two different panels does not match. Are the authors showing a parameter other than CD3 in the right panel?

6) The flow cytometric analysis of CamK4 expression shown in Fig. S1B and Fig. 1C-D is dubious, especially given that online gene expression repositories (such as Immgen or the Human Protein Atlas) indicate that monocytes do not express CamK4. Additional validation of this data has to be provided, either by proving the specificity of this antibody in cells where CaMK4 is knocked down, or by analysing transcript levels of Camk4 in FACS-sorted T cells and monocytes.

7) In the legend to Figure 3 it is stated that "data are representative of two independent experiments with 7-8 mice per group". Since only 3-4 symbols representing individual

mice are shown in Fig 3B-D, I assume that the authors did performed 2 experiments with 3-4 mice/group each. How was the statistics done?

8) According to Fig 1F, IMQ treatment induces upregulation of CAMK4 protein in different types of immune cells (except B cells) and KC. Specificity of the antibody used for Western blot should be confirmed by comparing sorted cells from WT and CamK4^{-/-} mice. Furthermore,

9) Further to Fig. 1F and G. MACS-based sorting of immune cells from the skin is usually not very pure. At a minimum, the purity after MACS should be determined and values provided in the figure legend. Using FACS-sorting would certainly be better.

10) Regarding the analysis of IL-10 expression by myeloid cells in the skin by flow cytometry (Fig. 4D) : as myeloid cells tend to be highly autofluorescent, it is crucial that the authors show a fluorescence-minus-one or an isotype control for IL-10.

REVIEWER COMMENTS

Reviewer #1 (Remarks to the Author):

This study represents an extension of studies which were reported previously and had established the role of CaMK4 in the expression of lupus, lupus nephritis and EAE.

1. Fig1. Neutrophils are known to be important in the skin pathology in psoriasis. The authors have ignored their presence. They should be included in this figure and in subsequent sets of data.

Response: Thanks for your important comments. According to your suggestion, we have sorted neutrophils of mouse skin and detected CaMK4 protein and gene levels of neutrophils (Supplementary Fig. 2). In subsequent sets of data, we have added the analysis of neutrophils.

2. Fig5. There are a number of issues. First KN93 inhibits phosphorylation of CaMK4 and not protein levels (one report, <https://bmcanesthesiol.biomedcentral.com/articles/10.1186/s12871-016-0191-4>).

Therefore, something is wrong here. The data here and in figure 7 are inferential. Production of IL10 should and be studied better. A major transcription factor for IL10 is STAT3 (PMID: 27885845) which is generated in response to IL23 and there is lots of IL-23 in the psoriatic skin lesions. This reviewer would like to understand why would IL23 signal differently in different myeloid cells present in the skin

Response: Thanks for your important comments. Whether KN93 inhibits phosphorylation of CaMK4 or protein level, we have verified it again. BMDMs were treated with 5 μ M and 10 μ M KN-93 for 6 h and 12 h. The results showed that CaMK4 protein level did decrease with the increase of KN-93 concentration (Supplementary Fig. 6b). This may be due to the long-time combination of KN-93 with CaMK4 results in degradation of CaMK4 protein. As for why we didn't check phosphorylation of CaMK4, the biggest reason is that well-known companies (such as Cell Signaling

Technology, Abcam, R&D systems, and Santa Cruze) don't have p-CaMK4 antibody against mouse. The report (<https://bmcanesthesiol.biomedcentral.com/articles/10.1186/s12871-016-0191-4>) used p-CaMK4 antibody that purchased from Santa Cruze, however, the antibody is not found on Santa Cruze website.

We do agree with the reviewer's comments that indeed STAT3 is a major transcription factor for IL-10 and IL-23/STAT3/IL-10 axis plays a critical role in the skin. We have detailly discussed this content in Discussion section.

3. Fig6. Interestingly, they show that *Camk4*-KO and KN93-treated mice have decreased skin scaling and thickness (keratinocyte activation) BUT NOT erythema (local inflammation). However, when they treat mice with IL-10 (or inject an important quantity of *Camk4*-KO macrophage) there is a decrease in both skin scaling/thickness AND erythema. If IL-10 was mediated the effect in *Camk4*-KO/KN93-treated mice, we would expect to see a decrease in erythema in these.

The authors might want to confirm that it's macrophage IL-10 which mediates improvement of the disease (eg, use myeloid cell-specific IL10-KO).

Response: Thanks for your important comments. We did observe only the decrease in scale and thickness, but no change in erythema in *Camk4*-KO and KN93-treated mice compared to control mice. Consistent with this was that isolated immune cells was the same between *Camk4*-KO/KN93-treated mice and control mice.

Indeed it is better to use myeloid cell-specific *Il10*-KO mice to confirm that macrophage-derived IL-10 mediates improvement of the disease. But regretfully, we haven't got the myeloid cell-specific *Il10*-KO mice, we are sorry for this. As a substitution, we have used IL-10 blocking antibody to treat IMQ-induced *Camk4*-KO mice. As shown in Fig. 6l-o, we observed that the severity of psoriasis in anti-IL-10-treated mice was restored than that in control mice. Also, Seon-Pil Jin et al. found that Imiquimod-applied IL-10 deficient mice showed more persistent psoriasis-like inflammation and higher severity index than did WT mice (Exp Dermatol, 2018, 27:43-49).

4. KN93 inhibits both CaMK2 and 4. Have the authors checked the levels on CaMK2?

Response: Thanks for your important comments. We have added *in vivo* animal experiment and *in vitro* cell experiment to explain the specificity of KN-93. First, *in vivo* animal experiment, we detected *Camk2a*, *Camk2b*, *Camk2d*, and *Camk2g* mRNA levels of whole skin from IMQ-treated mice and control mice. *Camk2a* and *Camk2b* were decreased in the whole skin of IMQ-treated mice, whereas *Camk2d* and *Camk2g* had no changes (Supplementary Fig. 7). Compared to IMQ-induced *Camk4^{-/-}* mice, KN-93-treated IMQ-induced *Camk4^{-/-}* mice had no difference of IMQ-induced psoriatic symptoms (Supplementary Fig. 7). Second, *in vitro* cell experiment, we detected *Camk2a*, *Camk2b*, *Camk2d*, and *Camk2g* mRNA levels of BMDMs with or without IMQ treatment. Our results showed that *Camk2b* was upregulated in IMQ-treated BMDMs, whereas *Camk2a*, *Camk2d*, and *Camk2g* had no changes (Supplementary Fig. 6a). Indeed KN93 inhibits both CaMK2 and CaMK4, but there was no significant change in *Camk2* family genes in IMQ-treated mice and BMDMs, which indirectly prove that KN-93 inhibits CaMK4 to perform function.

5. In lupus, anti-CD3/TCR antibodies (ref 27) and possible autoantigen drive the phosphorylation of CaMK4. What drives the CaMK4 in psoriasis? Someone should assume TLR7 (the receptor for imiquimod). I wonder why the authors did not pursue this pathway.

Response: Thanks for your important comments. In lupus, CaMK4 was increased in the nucleus of SLE T cells and anti-CD3 antibody-treated normal T cells (ref 27: JCI, 2005, 115:996-1005). In the present study, we found that CaMK4 was increased in macrophages from IMQ-induced mice (Fig. 1f) and was also increased in BMDMs after IMQ stimulation (Fig. 5b). We can understand it as those results indicate that IMQ/TLR7 drives the CaMK4 in psoriasis.

Reviewer #2 (Remarks to the Author):

This is a paper by Liang Yong et al. on the possible role of Calcium/calmodulin-dependent protein kinase IV in psoriatic inflammation. First, the authors show by immunohistochemistry CaMK4 expression in psoriasis lesional skin. They also show that CaMK4 is present in peripheral blood mononuclear cells by flow cytometry. CaMK4 seems to be higher expressed in peripheral monocytes from psoriasis patients compared to healthy controls. In the psoriasis-like imiquimod (IMQ) mouse model, the authors found high expression of CaMK4, not only in T cells but also in F4/80+ cells, CD11c+ cells and keratinocytes. When using CaMK4 knockout mice, these mice developed reduced back skin thickness and scaling compared to wildtype mice. By RNA-seq analysis and quantitative PCR a reduction of S100a8, S100a9, Il1b, Il17a, Il17f, Il22, and Ccl20 was found in IMQ-treated Camk4 knockout mice. The authors then show that CaMK4 is also expressed in F4/80 and CD11c positive cells in the IMQ model and that the frequencies of DCs are decreased in Camk4 knockout mice. Finally, the authors provide data suggesting that CaMK4 in macrophages regulates IL-10 expression and that this regulation is mediated through the CaMK4 ADCY1 cAMP Erk1/2 and p38 pathways. For this, they used pharmacological inhibitors.

Although this paper is interesting, the authors do not provide a novel mechanism, which could be of therapeutic use. First, they do not provide evidence that CaMK4 specifically in macrophages regulates psoriatic inflammation through IL-10. For this, macrophages with genetic deletion in CaMK4, IL-10 or both would be helpful. Likewise, a transfer model could be helpful. Second, the results shown in the IMQ model may be very different from the situation in humans with psoriasis, where macrophages seem not to play a major role. Moreover, IL-10 administration was of no benefit in randomized and controlled human trials in psoriasis. If macrophages, IL-10 and cAMP are of importance in psoriasis, the authors should be able to find a regulation of these cellular and non-cellular factors in patients treated with apremilast.

The title of the paper is misleading. CaMK4 cannot mediate psoriasis inflammation.

Response: Thanks for your important comments. We do agree that using mice of macrophages with genetic deletion in CaMK4, IL-10 or both can provide direct evidence that CaMK4 specifically in macrophages regulates psoriatic inflammation

through IL-10. We are now generating the *Camk4*^{fl/fl} Lyz2-Cre mice, but regretfully, we haven't got the *Camk4*^{fl/fl} Lyz2-Cre mice because of the long breeding time, we are sorry for this. As a substitution, we have used IL-10 blocking antibody to treat IMQ-induced *Camk4*-KO mice. As shown in Fig. 6l-o, we observed that the severity of psoriasis in anti-IL-10-treated mice was restored than that in control mice. Also, Seon-Pil Jin et al. found that Imiquimod-applied IL-10 deficient mice showed more persistent psoriasis-like inflammation and higher severity index than did WT mice (Exp Dermatol, 2018, 27:43-49).

In human skin, a subpopulation of CD163-positive macrophages was classically activated in psoriatic lesions and produced inflammatory molecules (ref 58: JID, 2010, 130:2412-2422). However, the CD163 marker is more assimilated to the “alternative” macrophages or M2 in normal human skin. Macrophages in human skin may not have a single phenotype, M1 or M2, rather have a great phenotypic plasticity in responding to microenvironment. This is similar to IMQ-induced mouse model. The reports of macrophages in psoriasis of humans is much less, it is may due to macrophages is too fewer to isolate difficultly from human skin. Single-cell sequencing of human skin may make the function of macrophages clearer. We hope that our present study can provide a reference for further research of macrophages in psoriasis of humans.

Indeed IL-10 administration was performed at clinical trials in psoriasis with limited success, but yielded some promising results were undeniable. Further studies in understanding the regulation of IL-10 expression in a context-dependent manner will promote the development of autoimmune diseases. Deeper understanding of IL-10 biology is require, specifically the mechanisms of IL-10 production and action, as well as the contrasting roles IL-10 may play in different contexts.

We have added these two research defects into Discussion section. We thank you again for your valuable comments.

Psoriasis patients treated with apremilast, a PDE4 inhibitor, have increased levels of cAMP and IL-10, and have reversed features of the inflammatory pathophysiology in skin (Biochem Pharmacol, 2012, 83:1583-1590). This suggest that IL-10 inhibiting inflammatory response to alleviate psoriatic symptoms is a part of reason.

We have changed the title to “Calcium/calmodulin-dependent protein kinase IV mediates psoriasis-like inflammation via macrophages and keratinocytes in imiquimod-induced mice”.

Figs.1 and 2: The authors should provide higher resolution histology images.

Response: Thanks for your important comments. We have provided higher resolution histology images in Figure 1 and 2.

Fig.3: Flow cytometry dot plots of cytokine positive CD4⁺ T cells and $\gamma\delta$ T cells have to be added to this figure.

Response: Thanks for your important comments. We have added flow cytometry dot plots of cytokine-positive CD4⁺ T cells and $\gamma\delta$ T cells in Fig. 3.

Figs.3 and 4: The authors indicate that the data is from two experiments with 7 to 8 mice per group. The figures displayed show only 4 dots per group. Please show all 7 to 8 dots and describe the statistical test used in each figure legend.

Response: Thanks for your important comments. We are sorry for wrong writing. What we wanted to express was “data are representative from one of two independent experiments with 3-4 mice per group”. We have now combined and analyzed the results of two experiments. In addition, we have added the statistical test used in each figure legend.

Fig.5: In A the authors show beta actin as protein load, but the band is only visible in the input lane. Please explain or use a nuclear load control.

Response: Thanks for your important comments. In the co-IP assay, β -actin as protein load is thoughtless. In Fig. 5a and Fig. 7e, we have changed β -actin to CaMK4 as a load control.

Fig.7: The Western blot analysis depicted has different backgrounds. The authors should provide the original non-cropped blots.

Response: Thanks for your important comments. Due to the different exposure time, the background appears different. We have adjusted the background to be almost the same in Fig. 7f.

The statistics used in most of the figures seem not to be appropriate. For example the authors have groups of 3 or 4 data points and used a t-test, which requires Gaussian distribution.

Response: Thanks for your important comments. Indeed it is inappropriate and inaccurate to use a t-test statistically analyzing two groups of 3 or 4 data points. On the one hand, we have combined the results of two experiments and increased each group from 3-4 data points to 7-8 data points. On the other hand, the Gaussian distribution and variance homogeneity were measured then two groups of 7-8 data points were compared by the two-tailed unpaired Student's t-tests.

Reviewer #3 (Remarks to the Author):

The manuscript by Yong et al explores the function of CaMK4 in the development of psoriasis. They find that Camk4-deficient animals show slightly diminished severity of imiquimod (IMQ)-induced psoriasis. They suggest this be dependent on enhanced expression of IL-10 by Camk4-deficient myeloid cells and show that administration of IL-10 or of Camk4-deficient BMDMs can diminish the severity of IMQ-induced psoriasis. They also show that treatment of animals with KN-93, an inhibitor of CaM kinases, also diminishes severity of IMQ-induced psoriasis.

The phenotype of CamK4-deficient mice in IMQ-induced psoriasis is rather mild and not entirely surprising, considering the known involvement of CamK4 in IL-17A-mediated inflammatory pathologies. Also, the suggested mechanism is not convincingly demonstrated. On the other hand, the biochemical characterization of the function of CaMK4 and IL-10 regulation in BMDMs is quite interesting.

There are a number of important concerns that need to be addressed prior publication.

Main points:

1) The authors suggest that IL-10 upregulation in CaMK4-deficient animals is responsible for the reduced severity of IMQ-induced psoriasis, but do not formally prove it. In Fig. 7 they show that administration of IL-10 or injection of CamK4-deficient BMDMs diminishes the severity of psoriasis. However, in order to convincingly demonstrate that enhanced expression of IL-10 in the absence of CamK4 is responsible for the partial protection from psoriasis, they need to block IL-10 with a neutralizing antibody in CamK4-KO mice to see whether that restores normal severity to psoriasis.

Response: Thanks for your important comments. According to your suggestion, we have used IL-10 neutralizing antibody to treat IMQ-induced *Camk4*-KO mice, the results showed that the severity of psoriasis in anti-IL-10-treated mice was restored than that in control mice (Fig. 6l-o).

2) The characterization of the immune cells that infiltrate the skin upon IMQ-treatment by flow cytometry is rather poor. The main populations found in the lesional skin of IMQ-treated mice are not quantified. The gating strategy of myeloid cells shown in Fig. S3 appears inaccurate and incomplete. Langerhans cells are gated F4/80+ CD11c+ MHCII+, but in the inflamed skin of IMQ-treated mice this gate would probably also include monocyte-derived dendritic cells and/or monocyte-derived macrophages (moDCs). A langerin staining should be included or the claim should be rephrased.

Moreover, the flow cytometry analysis shown in Fig 3A and supp Fig 3. does not allow interpretation of the data and drawing the conclusions that the authors propose. If they want to highlight expansion or reduction of a certain population in the skin of CamK4-deficient animals, they should show the percentage of the indicated population among total CD45+ cells in the skin and total cell numbers (as it is back skin, probably the best way would be to show number of cells per gram of tissue). Otherwise, the values shown in Fig. 3A are meaningless and cannot be interpreted, unfortunately. Accordingly, the

statistical analysis in Fig 3E is meaningless.

Infiltration of T cells (ab and gd TCR) and neutrophils have to be quantified: percentages (among CD45⁺ cells) and total numbers of these cells in the skin need to be shown to allow a more accurate interpretation of the phenotype of Camk4-Ko mice. Surprisingly, looking at the dot plot, the number of neutrophils appear to be very low, which is contrast to many other reports.

Response: Thanks for your important comments. We do acknowledge the reviewer's comments that using F4/80, CD11c, MHC II, CD11b as the marker of Langerhans cells in the inflamed skin is inaccurate and incomplete. We have rephrased "LCs" to "F4/80⁺ CD11c⁺ cells" in Fig. 3 and Supplementary Fig. 4.

We have improved our flow cytometry data, including the percentage of the indicated population among total CD45⁺ cells in the skin and total cell number per cm² (Fig. 3) and reanalyzed correlation between changed cells (Supplementary Fig. 5).

Infiltration of T cells (CD4⁺ and $\gamma\delta$ TCR⁺) and neutrophils have been quantified in Fig. 3. In Supplementary Fig. 4a, the dot plot was analyzed from the untreated mice, so the number of neutrophils was very low.

3) The biochemical analysis performed in BMDM (Fig. 5) shows some of the more interesting data of the article. It suggests that CamK4 upregulation upon IMQ-treatment blocks IL-10 expression by inhibition of ADCY1/cAMP production and the activation of Erk and p38 in BMDM. However, the comparison of Fig. 5C and 5E shows that treatment of BMDMs with KN-93 has a more pronounced effect on IL-10 transcript expression than genetic ablation of CamK4. This is probably due to the fact that KN-93 inhibits not only CaMK4 but also other CaM kinases (such as CaMK1 and 2, Pellicena and Schulman, Front Pharmacol 2014) which probably also mediate part of the phenotype shown here. The broad specificity of KN-93 prompts important questions that need to be addressed: i) how is the expression of other CaM kinases affected by KN-93 treatment? ii) Does KN93-treatment affect psoriasis development in IMQ treated CaMK4^{-/-} mice? iii) how does the deficiency of CamK4 in BMDMs affect the expression of ADCY1, p-Erk, p-p38 and IL-10 (protein) upon IMQ treatment?

Response: Thanks for your important comments and explanation. To explain the broad specificity of KN-93, we have added *in vivo* animal experiment and *in vitro* cell experiment. First, *in vivo* animal experiment, we detected *Camk2a*, *Camk2b*, *Camk2d*, and *Camk2g* mRNA levels of whole skin from IMQ-treated mice and control mice. *Camk2a* and *Camk2b* were decreased in the whole skin of IMQ-treated mice, whereas *Camk2d* and *Camk2g* had no changes (Supplementary Fig. 7). Compared to IMQ-induced *Camk4*^{-/-} mice, KN-93-treated IMQ-induced *Camk4*^{-/-} mice had no difference of IMQ-induced psoriatic symptoms (Supplementary Fig. 7). Second, *in vitro* cell experiment, we detected *Camk2a*, *Camk2b*, *Camk2d*, and *Camk2g* mRNA levels of BMDMs with or without IMQ treatment. Our results showed that *Camk2b* was upregulated in IMQ-treated BMDMs, whereas *Camk2a*, *Camk2d*, and *Camk2g* had no changes (Supplementary Fig. 6a). Indeed KN93 inhibits both CaMK2 and CaMK4, but there was no significant change in *Camk2* family genes in IMQ-treated mice and BMDMs, which indirectly prove that KN-93 inhibits CaMK4 to perform function. In Fig. 5b (line 3 versus line 2) and 5c (column 3 versus column 2), ADCY1, p-Erk1/2, p-p38 and IL-10 was increased after CaMK4 inhibition. We speculate that CaMK4 may degrade ADCY1 through recruiting and phosphorylating ubiquitinase.

4) In Figure 7, the authors aim to determine the function of CamK4 in keratinocyte and to this end they transfect HaCaT cells with CaMK4 siRNA and observe mild effects on the expression of inflammatory mediators. However, the authors do not provide evidence of the efficiency of knock-down (i.e. Western Blot analysis of CaMK4 in siRNA and scrambled treated cells). Also, they depict a cell cycle analysis in Fig. 7C, but there is no information in the manuscript on how this was done. This should be rectified.

Response: Thanks for your important comments. The efficiency of siCAMK4 knockdown was shown in Fig. 7f. After CaMK4 knockdown, the band of CaMK4 protein became shallower than scrambled control.

In Methods section, we have added the description of cell apoptosis and cell cycle. Cell apoptosis and cell cycle were measured by commercially available kits according to the

manufacturer's instructions.

Minor points:

5) There is an inconsistency in the gating strategy shown in Fig S1B. On the left panel, DN T cells are shown to be gated as CD45⁺ CD3⁺ CD4⁻ CD8⁻ cells. However, on the right panel these cells are shown as CD3⁻. A closer look at the axes shows that the scale for CD3 in the two different panels does not match. Are the authors showing a parameter other than CD3 in the right panel?

Response: Thanks for your important comments. Through careful check, we found that we mistakenly wrote CD4 as CD3. Now we have corrected it.

6) The flow cytometric analysis of CamK4 expression shown in Fig. S1B and Fig. 1C-D is dubious, especially given that online gene expression repositories (such as Immgen or the Human Protein Atlas) indicate that monocytes do not express CamK4. Additional validation of this data has to be provided, either by proving the specificity of this antibody in cells where CaMK4 is knocked down, or by analysing transcript levels of Camk4 in FACS-sorted T cells and monocytes.

Response: Thanks for your important comments. We checked the expression of *CAMK4* on The Human Protein Atlas website and found that *CAMK4* is expressed in monocytes from single cell data of blood. The same CaMK4 antibody are used in Fig. S1B, Fig. 1C-D and Fig. 7f. In Fig. 7f, after CaMK4 knockdown, the band became shallower.

7) In the legend to Figure 3 it is stated that “data are representative of two independent experiments with 7-8 mice per group”. Since only 3-4 symbols representing individual mice are shown in Fig 3B-D, I assume that the authors did performed 2 experiments with 3-4 mice/group each. How was the statistics done?

Response: Thanks for your important comments. We are sorry for wrong writing. What we wanted to express was “data are representative from one of two independent experiments with 3-4 mice per group”. We have now combined and analyzed the results

of two experiments.

8) According to Fig 1F, IMQ treatment induces upregulation of CAMK4 protein in different types of immune cells (except B cells) and KC. Specificity of the antibody used for Western blot should be confirmed by comparing sorted cells from WT and CamK4^{-/-} mice.

Response: Thanks for your important comments. Specificity of the CaMK4 antibody used for Western blot have been confirmed in Supplementary Fig. 2a.

9) Further to Fig. 1F and G. MACS-based sorting of immune cells from the skin is usually not very pure. At a minimum, the purity after MACS should be determined and values provided in the figure legend. Using FACS-sorting would certainly be better.

Response: Thanks for your important comments. After MACS, the purity of sorted cells was determined by a flow cytometer and the purity was both > 90%. In Methods section, we mentioned “The purity of sorted cells was > 90%”. Now we have provided it in the figure legend.

10) Regarding the analysis of IL-10 expression by myeloid cells in the skin by flow cytometry (Fig. 4D): as myeloid cells tend to be highly autofluorescent, it is crucial that the authors show a fluorescence-minus-one or an isotype control for IL-10.

Response: Thanks for your important comments. We have added isotype control for IL-10 in Fig. 4d.

REVIEWER COMMENTS

Reviewer #1 (Remarks to the Author):

The manuscript has been improved.

The title as edited is not grammatically correct.

Abstract "imiquimod (IMQ)-induced.." induced to what?

The abstract needs to be presented better. As it is, it reads disjointed.

Still, there is no link between TLR7 and CaMK4 expression.

Notwithstanding the difficulty with KO mice, the conclusions remain not conclusive.

The text needs to be brought to style e.g. "..mice back skin.". "Camk4 deficiency mice...." lines 405-408 need rephrasing, and so on. All new text in the discussion needs to be rephrased. There are too many stylistic changes that need to be done.

The cartoon in Fig. 9, does not include data presented in Figure 7.

The use of UO126 is not discussed in the main text or the figure legend.

Reviewer #2 (Remarks to the Author):

This is a revised paper by Liang Yong et al. on the possible role of Calcium/calmodulin-dependent protein kinase IV in psoriatic inflammation.

The revision and the rebuttal letter are written in poor quality. It is unpleasant to read and hard to understand such a low quality rebuttal letter.

Title – what are imiquimod-induced mice?

They did not answer the key point raised by referee #1, point 3 on the use of myeloid cell-specific IL10-KO mice. Similarly, referee #2 raised the same point to provide evidence that CaMK4 specifically in macrophages regulates psoriatic inflammation through IL-10. This is a very critical experiment for the whole paper. IL-10 blocking antibody is of no value in this experimental setting.

They did not convincingly answer the questions 4 and 5 of referee #1. They also did not respond to the concerns raised by referee #2.

The flow cytometry data presented assume that the techniques are not well performed. CD4 staining and cytokine staining looks odd. The statistical tests used are still not correct.

It seems that the original submission contained many inattentions and the authors' reply shows that they seem not to take these scientific points very serious.

Reviewer #3 (Remarks to the Author):

The authors have reasonably addressed the majority of questions

Reviewer #1 (Remarks to the Author):

The manuscript has been improved.

Response: Thanks for your recognition. We would like to thank you for spending your valuable time to review manuscript.

The title as edited is not grammatically correct.

Response: We would like to thank you for this important comment. We are sorry for that we abruptly added “in imiquimod-induced mice” at the end of the original title “Calcium/calmodulin-dependent protein kinase IV mediates psoriasis-like inflammation via macrophages and keratinocytes”. It seems to be inappropriate. We have now changed the title to “Calcium/calmodulin-dependent protein kinase IV mediates imiquimod-induced psoriatic inflammation via macrophages and keratinocytes” and we hope that it is now correct.

Abstract “imiquimod (IMQ)-induced..” induced to what?

Response: We would like to thank for your important comments. We apologize for using this wrong phrase. We have changed “IMQ-induced mice” to “IMQ-treated mice” and corrected all of the same mistakes in the manuscript.

The abstract needs to be presented better. As it is, it reads disjointed.

Response: We would like to thank for your important comments. We read the Abstract and indeed felt it disjointed. We have improved the Abstract to make it reading jointed.

Still, there is no link between TLR7 and CaMK4 expression.

Response: We are sorry that we haven't solved this key point in the last revision. We would like to thank you for raising again this important question to improve our manuscript.

Because the commercially available p-CaMK4 antibody is just against human, now we select human monocyte line THP-1 cells as the research object.

Methods: 5×10^5 or 1.5×10^6 THP-1 cells suspended in 1 ml culture medium were seeded in a 24-well plate or a 12-well plate, respectively, and stimulated with PMA (100 ng/ml) for 24 h to differentiate into macrophages (THP-1-macrophages) for subsequent experiments. The cells were treated with IMQ (2 μ g/ml) for 1 h, 2 h, 3 h, 6 h, 12 h, and 24 h to extract RNA and protein. For *TLR7* knockdown, THP-1-macrophages were transfected with siRNA targeting *TLR7* (si*TLR7*-1: 5'-GCUCAAAUCUUUCAGUUGGTT-3', si*TLR7*-2: 5'-CCUGUGAGUUAGAUCUGACTT-3', and si*TLR7*-3: 5'-UCAGGAGUCUGACGAAGUATT-3') or scrambled control (5'-UUCUCCGAACGUGUCACGUTT-3') for 24 h using lipofectamine 3000. Then cells were treated with IMQ (2 μ g/ml) for 24 h. The cells were harvest for RNA and protein extractions.

Results: We found that *CAMK4* mRNA levels were upregulated at 1 h, 2 h, 3 h, 6 h, and 24 h after IMQ treatment compared to controls (Response letter Fig. 1a). The protein levels of CaMK4 were increased at 12 h and 24 h after IMQ treatment, whereas phosphorylation of CaMK4 that relative to total protein of CaMK4 had no changes (Response letter Fig. 1b). TLR7 recognize its ligands, such as ssRNA and IMQ, to induce type I IFN transcription by IRF7 or to activate transcription factor NF- κ B^{1,2}. Our results showed that IMQ induced phosphorylation of NF- κ B p65 at 12 h and 24 h (Response letter Fig. 1b). These data preliminarily suggest that IMQ-TLR7 induces CaMK4 expression by the NF- κ B pathway. To further confirm this conclusion, we used siRNA to knockdown *TLR7* followed with IMQ treatment in THP-1-macrophages. Quantitative PCR and Western blot analyses provided evidence that si*TLR7*-3 had higher efficiency of knockdown and was suitable for subsequent experiments (Response letter Fig. 1c and 1d). After *TLR7* knockdown, CaMK4 mRNA and protein levels and phosphorylation of NF- κ B p65 were reduced in IMQ-treated cells compared to scrambled control-transfected ones (Response letter Fig. 1e and 1f). Taken together, these results indicate that the IMQ-TLR7-NF- κ B pathway drives CaMK4 expression.

Response letter Figure 1. The IMQ-TLR7-NF-κB pathway induces CaMK4 expression in THP-1-macrophages. a *CAMK4* expression at the indicated time points. **b** Western blot analysis of p-CaMK4, CaMK4, p-NF-κB p65, and NF-κB p65. **c** Quantitative PCR analysis of *TLR7* after siRNAs transfection. SC, scrambled control. **d** Western blot analysis of TLR7. **e** mRNA expression of *CAMK4* as indicated. **f** Western blot analysis of p-CaMK4, CaMK4, p-NF-κB p65, and NF-κB p65 as indicated.

Notwithstanding the difficulty with KO mice, the conclusions remain not conclusive.

Response: We would like to thank for your important comments. Firstly, we apologize for not getting the myeloid cell-specific *Camk4* KO mice in the last revision. This time, we have got the myeloid cell-specific *Camk4* KO mice (*Camk4^{fl/fl} Lyz2-Cre* mice) and further verified our results. As shown in Fig. 7, *Camk4^{fl/fl} Lyz2-Cre* mice treated with IMQ showed more extenuative psoriatic phenotypes than control mice.

The text needs to be brought to style e.g. “..mice back skin..” “*Camk4* deficiency mice....” lines 405-408 need rephrasing, and so on. All new text in the discussion needs

to be rephrased. There are too many stylistic changes that need to be done.

Response: We would like to thank for your important comments. We have rephrased the texts, such as “mice back skin”, “Camk4 deficiency mice”, and all new text in the Discussion. To improve the language presentation and quality of our manuscript, we have carried out editing service at the AJE website.

The cartoon in Fig. 9, does not include data presented in Figure 7.

Response: We would like to thank for your important comments. We have added the data presented in Fig. 7 (now Fig. 8) into the cartoon in Fig. 9 (now Fig. 10). Apologetically, we think there are something wrong in Fig. 10. We have updated the cartoon and we hope that it is now correct.

The use of UO126 is not discussed in the main text or the figure legend.

Response: We would like to thank for your important comments. We have added the use of UO126 in the section “Methods-MACS sorting and cell treatment”.

Reviewer #2 (Remarks to the Author):

This is a revised paper by Liang Yong et al. on the possible role of Calcium/calmodulin-dependent protein kinase IV in psoriatic inflammation.

The revision and the rebuttal letter are written in poor quality. It is unpleasant to read and hard to understand such a low quality rebuttal letter.

Response: First of all, we are sincerely sorry for our poor-revised manuscript and low-quality rebuttal letter. Please forgive our many mistakes. We would like to thank you for spending your valuable time to review our poor-revised manuscript. We will capture this valuable opportunity and go all out to revise the manuscript seriously according to all of the concerns raised by referees. Again, we are sorry for our stiff phrases.

Title – what are imiquimod-induced mice?

Response: We would like to thank for your important comments. We apologize for using this wrong phrase. We have now changed the title to “Calcium/calmodulin-dependent protein kinase IV mediates imiquimod-induced psoriatic inflammation via macrophages and keratinocytes” and we hope that it is now correct.

They did not answer the key point raised by referee #1, point 3 on the use of myeloid cell-specific IL10-KO mice. Similarly, referee #2 raised the same point to provide evidence that CaMK4 specifically in macrophages regulates psoriatic inflammation through IL-10. This is a very critical experiment for the whole paper. IL-10 blocking antibody is of no value in this experimental setting.

Response: We are sorry for that we haven't got the myeloid cell-specific *Camk4* or *Il10* KO mice to solve this key point in the last revision. We appreciate that you and Reviewer #1 mention this important issue in two revisions to help us improving our paper. This time, we have got the myeloid cell-specific *Camk4* KO mice (*Camk4^{fl/fl} Lyz2-Cre* mice) and further verified our results. As shown in Fig. 7, *Camk4^{fl/fl} Lyz2-Cre* mice treated with IMQ showed more extenuative psoriatic phenotypes than control mice.

They did not convincingly answer the questions 4 and 5 of referee #1. They also did not respond to the concerns raised by referee #2.

Response: We are sorry that we haven't solved these important questions well in the last revision. We would like to thank you for raising again these key points to improve our manuscript.

For the question 4 (KN93 inhibits both CaMK2 and 4. Have the authors checked the levels on CaMK2?) of referee #1, we referred to Tomohiro Koga's research³ and detected the expression of *Camk4* and *Camk2* family genes upon IMQ treatment. The results showed that CaMK4 was preferentially induced; thus, we conjecture that KN-93 inhibits CaMK4 to perform function upon IMQ treatment.

For the question 5 (What drives the CaMK4 in psoriasis? Someone should assume TLR7 (the receptor for imiquimod)) of referee #1, we have solved it. As shown in

Response letter Fig. 1, we found that the IMQ-TLR7-NF- κ B pathway induces CaMK4 expression.

For the concerns raised by referee #2, we sincerely apologize for our stiff wording and inaccurate response in the last revision. We have rephrased the texts added in the Discussion to be aware of the deficiency of our paper. We hope this will satisfy your opinion and hope this is appropriate. We would like to thank you again.

The flow cytometry data presented assume that the techniques are not well performed. CD4 staining and cytokine staining looks odd. The statistical tests used are still not correct.

Response: We would like to thank for your important comments. We are sorry for the odd CD4 staining and cytokine staining and wrong statistical tests.

In the original Fig. 3c, CD4⁺ cells look diffuse and not clustered. To solve this problem, we have repeated the flow cytometry and the data is presented in the new Fig. 3.

For the statistical tests, we think the description is undeniably leaky. We have checked all of the statistical tests and rephrased the section “Methods-Statistics”. For comparing variables between two groups, if the data were followed Gaussian distribution, two-tailed unpaired Student’s *t*-test was used; if not, Mann-Whitney test was used.

It seems that the original submission contained many inattentions and the authors' reply shows that they seem not to take these scientific points very serious.

Response: We are sincerely sorry for our unserious attitudes, stiff phrases, poor-revised manuscript, and low-quality rebuttal letter. We wanted to do it well, but we didn't handle it well in the last revision. This time, we will try our best to do better. We are sorry again.

Reviewer #3 (Remarks to the Author):

The authors have reasonably addressed the majority of questions

Response: We would like to thank for your recognition. We appreciate your valuable comments for improving our manuscript.

References

- 1 Eng, H. L., Hsu, Y. Y. & Lin, T. M. Differences in TLR7/8 activation between monocytes and macrophages. *Biochemical and biophysical research communications* **497**, 319-325, doi:10.1016/j.bbrc.2018.02.079 (2018).
- 2 Diebold, S. S., Kaisho, T., Hemmi, H., Akira, S. & Reis e Sousa, C. Innate antiviral responses by means of TLR7-mediated recognition of single-stranded RNA. *Science* **303**, 1529-1531, doi:10.1126/science.1093616 (2004).
- 3 Koga, T. *et al.* CaMK4-dependent activation of AKT/mTOR and CREM-alpha underlies autoimmunity-associated Th17 imbalance. *The Journal of clinical investigation* **124**, 2234-2245, doi:10.1172/JCI73411 (2014).

REVIEWER COMMENTS

Reviewer #1 (Remarks to the Author):

The authors have addressed the raised concerns. Yet, a few more remain open, and more importantly, reading this manuscript becomes a painful exercise.

Abstract is still poor and difficult to understand.

"In the skin of IMQ-treated Camk4+/+ and Camk4^{-/-} mice, increased macrophages had a negative linear correlation with decreased IL-17A+ $\gamma\delta$ TCR+ 8 cells" not clear.

"CaMK4 functioned in macrophages by reducing IL-10 production, thus dysregulating the limitation on excessive psoriatic inflammation" torturous expression.

"Furthermore, myeloid cell-specific Camk4 knockout mice treated with IMQ also showed more extensive psoriatic phenotypes than control mice" what do they want to say?

Actually, all sentences need improvement.

L. 101: CaMK4-positive 101 signals ?? be exact. What signals?

Fig. 1. I know it was not mentioned before but for the human skin staining experiments, non-involved skin from the same patient is needed. It is quite possible that non-lesioned skin from patients with psoriatic arthritis, display increased amounts of CaMK4. Ed Vital in a recent Nature Communications paper showed abnormalities in non-lesioned skin from lupus patients.

L.119: "...and the mechanism of CaMK4 in T cells is clear^{24,27}" how is it clear, what are you referring to.

Too many grammatically or syntactically or not clear sentences to list them here. The authors can use an English medical writing service to bring the paper to a proper style.

L.200 "CaMK4 inhibits macrophages producing IL-10 .." do you mean that CaMK4 inhibits the production of IL-10 by macrophages???

"Partial proteins binding to CaMK4..." do you mean partial list of proteins?

Fig 5a. the reverse co-IP is needed.

L 1072 and elsewhere the word extenuative is not properly used.

7f: the open circles are denoted as closed circles in the legend.

Reviewer #2 (Remarks to the Author):

Yong et al. present a second revision of their manuscript on the role of calcium/calmodulin protein kinase IV in imiquimod-induced skin inflammation.

Title should be change, a protein kinase can not 'mediate' inflammation.

The manuscript text is still in poor quality and the abstract doesn't read well.

Abstract First sentence starts with psoriasis associated genes and then talks about CAMK4 protein. There is no relation of these two statements. Second sentence describe significantly increased CAMK4 in skin and mice – but compared to what? It is unclear what exactly is meant by 'significantly increased'. Sentence no 4: What is the meaning of 'increased macrophages had a negative linear correlation with decreased IL-17+ $\gamma\delta$ TCR+ cells'? Cell numbers? Please rewrite the abstract.

Results – the authors write that CAMK4 mRNA levels are increased in psoriasis patients' peripheral blood compared to HC. Why is CAMK4 mRNA secreted and circulates in the blood?

The authors correlated CAMK4 expression with levels of IL1B and IL12B. Why did they choose these factors and did not correlated CAMK4 with levels of IL17A and IL23A, which are more important in psoriasis pathogenesis than IL1B and IL12B?

Why did the authors not found significant changes in Il10 when performing their RNAseq analysis in Camk4^{-/-} mice treated with IMQ?

How do the authors explain the incredible high number of IL-4 positive T cells in Figure 3d? What is the difference between the figures 962 and 963?

Discussion

The wording should be changed. The authors did not study T17 differentiation. Therefore they should avoid statements on the role of CaMK4 in T17 cell differentiation. Their observations are based on the promotion of T17 responses in vivo in the IMQ model.

Taken together the authors improved their manuscript by adding new data, especially by performing experiments with the conditional ko mice. Yet, the text need much more improvement.

Reviewer #1 (Remarks to the Author):

The authors have addressed the raised concerns. Yet, a few more remain open, and more importantly, reading this manuscript becomes a painful exercise.

Abstract is still poor and difficult to understand.

“In the skin of IMQ-treated *Camk4*^{+/+} and *Camk4*^{-/-} mice, increased macrophages had a negative linear correlation with decreased IL-17A⁺ $\gamma\delta$ TCR⁺ cells” not clear.

“CaMK4 functioned in macrophages by reducing IL-10 production, thus dysregulating the limitation on excessive psoriatic inflammation” torturous expression.

“Furthermore, myeloid cell-specific *Camk4* knockout mice treated with IMQ also showed more extenuative psoriatic phenotypes than control mice” what do they want to say?

Actually, all sentences need improvement.

Response: We would like to thank you for your important comments. On the basis of your great suggestion, we have revised the sentences listed here and have rewritten the Abstract. The Abstract is presented below.

Abstract - CaMK4 plays a critical role in autoimmune diseases, and the role of CaMK4 in psoriasis remains obscure. Here, we show that CaMK4 expression is significantly increased in psoriatic lesional skin from psoriasis patients compared to healthy human skin as well as from imiquimod (IMQ)-induced psoriatic mice compared to healthy mouse skin. *Camk4*-deficient (*Camk4*^{-/-}) mice treated with IMQ exhibit reduced severity of psoriasis compared to wild-type (WT) mice. There are more macrophages and fewer IL-17A⁺ $\gamma\delta$ TCR⁺ cells in the skin of IMQ-treated *Camk4*^{-/-} mice compared to IMQ-treated WT mice. CaMK4 inhibits IL-10 production by macrophages, thus allowing excessive psoriatic inflammation. Correspondingly, loss of *Camk4* in macrophages alleviates IMQ-induced psoriatic inflammation. In keratinocytes, CaMK4 inhibits apoptosis as well as promotes cell proliferation and the expression of pro-inflammatory genes (*S100A8*, *CAMP*, etc.). Taken together, our data reveal that CaMK4 contributes to IMQ-induced psoriasis by sustaining inflammation and provide potential targets for psoriasis treatment.

l.101: CaMK4-positive signals?? be exact. What signals?

Fig. 1. I know it was not mentioned before but for the human skin staining experiments, non-involved skin from the same patient is needed. It is quite possible that non-lesioned skin from patients with psoriatic arthritis, display increased amounts of CaMK4. Ed Vital in a recent Nature Communications paper showed abnormalities in non-lesioned skin from lupus patients.

Response: We would like to thank you for this important and constructive comment. We have revised “CaMK4-positive signals” to “the number of CaMK4⁺ cells” in the text.

For the human skin staining experiments, we have added CaMK4 staining of paired psoriatic lesional skin and non-lesional skin in Supplementary Fig. 1a. As you predicted, an increased amount of CaMK4 was observed in psoriatic non-lesional skin compared to healthy skin.

The description in the text is “Immunohistochemical analysis of human skin tissues showed that the number of CaMK4⁺ cells in both the epidermis and dermis of psoriatic lesional skin was higher than that of non-lesional skin and healthy skin (Fig. 1a and Supplementary Fig. 1a). Meanwhile, an increased number of CaMK4⁺ cells in the epidermis and dermis of psoriatic non-lesional skin was observed compared to healthy skin (Supplementary Fig. 1a)”.

l.119: “...and the mechanism of CaMK4 in T cells is clear^{24,27}” how is it clear, what are you referring to.

Response: We would like to thank you for this important comment. We have described the mechanism of CaMK4 in T cells and have revised the sentence to “Activated CaMK4 mitigates IL-2 transcription by phosphorylating CREM- α , whereas it promotes IL-17 production through the AKT/mTOR/S6K/ROR γ t and CREM- α pathways in T cells of SLE”.

Too many grammatically or syntactically or not clear sentences to list them here. The authors can use an English medical writing service to bring the paper to a proper style.

1.200 “CaMK4 inhibits macrophages producing IL-10..” do you mean that CaMK4 inhibits the production of IL-10 by macrophages???

“Partial proteins binding to CaMK4...” do you mean partial list of proteins?

Response: We would like to thank you for your great comments. We have revised the sentences as follows:

The sentence “CaMK4 inhibits macrophages producing IL-10 via the ADCY1-cAMP-Erk1/2 and p38 pathways to regulate IMQ-induced psoriasis” has been revised to “CaMK4 inhibits IL-10 production by macrophages through the ADCY1-cAMP-Erk1/2 and p38 pathways to promote IMQ-induced psoriasis”.

The sentence “Partial proteins binding to CaMK4 were listed in Supplementary Table 1” has been revised to “A partial list of proteins binding to CaMK4 was shown in Supplementary Table 1”.

Additionally, a native English-speaking and professional scientific editor at Write Science Right has edited the manuscript.

Fig 5a. the reverse co-IP is needed.

Response: We would like to thank you for this important comment. We have performed a reverse co-IP assay and added the results to Fig. 5a. The results of the reverse co-IP assay are consistent with those of the forward co-IP assay.

1.1072 and elsewhere the word extenuative is not properly used.

Response: We would like to thank you for this important comment. We apologize for using this inappropriate word. We have deleted the word extenuative and have rephrased the sentences containing the word extenuative.

7f: the open circles are denoted as closed circles in the legend.

Response: We would like to thank you for this important comment. We have corrected the mistake and have revised the legend of Fig. 7f.

Reviewer #2 (Remarks to the Author):

Yong et al. present a second revision of their manuscript on the role of calcium/calmodulin protein kinase IV in imiquimod-induced skin inflammation.

Title should be change, a protein kinase can not 'mediate' inflammation.

Response: We would like to thank you for this important comment. We have changed the title to "Calcium/calmodulin-dependent protein kinase IV promotes imiquimod-induced psoriatic inflammation via macrophages and keratinocytes".

The manuscript text is still in poor quality and the abstract doesn't read well.

Response: We would like to thank you for this important comment. A native English-speaking and professional scientific editor at Write Science Right has edited the manuscript.

Abstract

First sentence starts with psoriasis associated genes and then talks about CAMK4 protein. There is no relation of these two statements.

Second sentence describe significantly increased CAMK4 in skin and mice-but compared to what? It is unclear what exactly is meant by 'significantly increased'.

Sentence no 4: What is the meaning of 'increased macrophages had a negative linear correlation with decreased IL-17⁺ gd TCR⁺ cells'? Cell numbers?

Please rewrite the abstract.

Response: We would like to thank you for your important comments. On the basis of your great suggestion, we have revised the sentences listed here and have rewritten the Abstract. The Abstract is presented below.

Abstract - CaMK4 plays a critical role in autoimmune diseases, and the role of CaMK4 in psoriasis remains obscure. Here, we show that CaMK4 expression is significantly increased in psoriatic lesional skin from psoriasis patients compared to healthy human skin as well as from imiquimod (IMQ)-induced psoriatic mice compared to healthy mouse skin. *Camk4*-deficient (*Camk4*^{-/-}) mice treated with IMQ exhibit reduced

severity of psoriasis compared to wild-type (WT) mice. There are more macrophages and fewer IL-17A⁺ $\gamma\delta$ TCR⁺ cells in the skin of IMQ-treated *Camk4*^{-/-} mice compared to IMQ-treated WT mice. CaMK4 inhibits IL-10 production by macrophages, thus allowing excessive psoriatic inflammation. Correspondingly, loss of *Camk4* in macrophages alleviates IMQ-induced psoriatic inflammation. In keratinocytes, CaMK4 inhibits apoptosis as well as promotes cell proliferation and the expression of pro-inflammatory genes (*S100A8*, *CAMP*, etc.). Taken together, our data reveal that CaMK4 contributes to IMQ-induced psoriasis by sustaining inflammation and provide potential targets for psoriasis treatment.

Results

The authors write that CAMK4 mRNA levels are increased in psoriasis patients' peripheral blood compared to HC. Why is CAMK4 mRNA secreted and circulates in the blood?

Response: We apologize for our incorrect description. We detected the mRNA level of *CAMK4* in the cells of peripheral blood. What we meant to say "Compared to healthy controls, the mRNA level of *CAMK4* in the cells of peripheral blood was also increased in patients with psoriasis". We have revised this sentence in the text.

The authors correlated CAMK4 expression with levels of IL1B and IL12B. Why did they choose these factors and did not correlated CAMK4 with levels of IL17A and IL23A, which are more important in psoriasis pathogenesis than IL1B and IL12B?

Response: We would like to thank you for this important comment. Indeed, we did look for the expression of *IL17A* and *IL23A* in the cells of peripheral blood from healthy controls and patients with psoriasis. There was no significant difference in the expression of *IL17A* and *IL23A* between two groups; thus, we did not analyze the correlation of *CAMK4* expression with the expression of *IL17A* and *IL23A*. Although there was no significant difference in the expression of *IL17A* between the cells of peripheral blood from healthy controls and patients with psoriasis, CaMK4 has been reported to regulate the production of IL-17A. Our results also found that CaMK4

promoted IL-17A production by Th17 cells and $\gamma\delta$ T17 cells in the skin of IMQ-treated mice. In addition, the cytokine IL-12/23p40 (*IL12B*) is also an important and effective therapeutic target for psoriasis.

Why did the authors not find significant changes in *Il10* when performing their RNAseq analysis in *Camk4*^{-/-} mice treated with IMQ?

Response: We would like to thank you for this important comment. The sample size used for RNA-seq was small (3 mice per group), and *Il10* expression had an increasing trend in the skin of IMQ-treated *Camk4*^{-/-} mice compared to IMQ-treated *Camk4*^{+/+} mice ($p = 0.0664$). Thus, we detected *Il10* expression using a larger sample size (8 mice per group) and found that *Il10* expression in the skin of IMQ-treated *Camk4*^{-/-} mice was significantly higher than that in IMQ-treated *Camk4*^{+/+} mice ($p = 0.0192$) (Fig. 4a).

How do the authors explain the incredible high number of IL-4 positive T cells in Figure 3d? What is the difference between the figures 962 and 963?

Response: We would like to thank you for this important comment. In Fig. 3d of revision 2 (Figure 962), we did not perform the flow cytometry technique and adjust the compensation well, thus leading to the incredible high number of IL-4-positive T cells. After your comment about the odd CD4 and cytokine staining and the poorly performed flow cytometry technique, we repeated the flow cytometry experiment and readjusted the compensation. In revision 3 (Figure 963), we corrected the inappropriate result about the incredible high number of IL-4-positive T cells. We thank you for your valuable comments again!

Discussion

The wording should be changed. The authors did not study T17 differentiation. Therefore they should avoid statements on the role of CaMK4 in T17 cell differentiation. Their observations are based on the promotion of T17 responses in vivo in the IMQ model.

Response: We would like to thank you for your important and constructive comments.

We agree with you and have revised the sentence “In the present study, we found that CaMK4 was not only required for Th17 cell differentiation, but also necessary specifically for $\gamma\delta$ T17 cell differentiation in mouse skin, which was consistent with the finding that dominant $\gamma\delta$ T17 cells were most strongly affected by *Camk4* deficiency” to “In the present study, we found that CaMK4 promoted IL-17A production by Th17 cells and $\gamma\delta$ T17 cells and that IL-17A was mainly produced by $\gamma\delta$ T cells in the skin of IMQ-treated mice”.

Taken together the authors improved their manuscript by adding new data, especially by performing experiments with the conditional ko mice. Yet, the text need much more improvement.

Response: We would like to thank you for your positive comments. We have improved the text. We appreciate your help with improving our paper.

REVIEWERS' COMMENTS

Reviewer #1 (Remarks to the Author):

the authors have improved significantly the presentation. Still many sentences need rewriting for clarity and style

Reviewer #2 (Remarks to the Author):

Liang Yong and colleagues present a second revision of their manuscript on the role of Ca/calmodulin protein kinase IV in IMQ-induced psoriatic inflammation. They studied a mechanism that links Ca/calmodulin protein kinase IV to macrophages, IL-10 expression and its effects on psoriatic inflammation in a mouse model.

Taken together the paper improved by the two rounds of major revision. Yet, the abstract and some parts of the text still need improvement by a native speaker. The findings are of interest and in total the paper is in much better shape now.

Reviewer #1 (Remarks to the Author):

The authors have improved significantly the presentation. Still many sentences need rewriting for clarity and style.

Response: We would like to thank you for this important comment. We appreciate your valuable guidance and we are deeply thankful for your help with improving our paper over the past year. For the language issues, a native English-speaking and professional scientific editor at AJE has edited the manuscript again.

Reviewer #2 (Remarks to the Author):

Liang Yong and colleagues present a second revision of their manuscript on the role of Ca/calmodulin protein kinase IV in IMQ-induced psoriatic inflammation. They studied a mechanism that links Ca/calmodulin protein kinase IV to macrophages, IL-10 expression and its effects on psoriatic inflammation in a mouse model.

Taken together the paper improved by the two rounds of major revision. Yet, the abstract and some parts of the text still need improvement by a native speaker. The findings are of interest and in total the paper is in much better shape now.

Response: We would like to thank you for your positive comments. We appreciate your help with improving our paper and we deeply feel grateful for your valuable time in guiding our study over the past year. For the language issues, a native English-speaking and professional scientific editor at AJE has edited the manuscript again.